# HNF4A defines tissue-specific circadian rhythms by beaconing BMAL1::CLOCK chromatin binding and shaping the rhythmic chromatin landscape

Meng Qu 1,4✉, Han Qu 2,4, Zhenyu Jia2,3 & Steve A. Kay 1✉

Transcription modulated by the circadian clock is diverse across cell types, underlying circadian control of peripheral metabolism and its observed perturbation in human diseases. We report that knockout of the lineage-specifying *Hnf4a* gene in mouse liver causes associated reductions in the genome-wide distribution of core clock component BMAL1 and accessible chromatin marks (H3K4me1 and H3K27ac). Ectopically expressing HNF4A remodels chromatin landscape and nucleates distinct tissue-specific BMAL1 chromatin binding events, predominantly in enhancer regions. Circadian rhythms are disturbed in *Hnf4a* knockout liver and HNF4A-MODY diabetic model cells. Additionally, the epigenetic state and accessibility of the liver genome dynamically change throughout the day, synchronized with chromatin occupancy of HNF4A and clustered expression of circadian outputs. Lastly, *Bmal1* knockout attenuates HNF4A genome-wide binding in the liver, likely due to downregulated *Hnf4a* transcription. Our results may provide a general mechanism for establishing circadian rhythm heterogeneity during development and disease progression, governed by chromatin structure.

¹ Department of Neurology, Keck School of Medicine, University of Southern California, Los Angeles, CA 90089, USA. ² Department of Botany and Plant Sciences, University of California, Riverside, CA 92521, USA. ³ Graduate Program in Genetics, Genomics, and Bioinformatics, University of California, Riverside, CA 92521, USA. ⁴These authors contributed equally: Meng Qu, Han Qu. ✉email: mengqu@usc.edu; stevekay@usc.edu

The circadian clock is a molecular oscillator that aligns behavior and physiology with daily light–dark cycles. The core of the mammalian circadian clock, composed of two interlocked transcriptional feedback loops, relies on chromatin occupancy of the master transcription factor heterodimer BMAL1::CLOCK at the E-box DNA element. BMAL1::CLOCK positively regulates expression of the *Period* (*Per1, Per2, Per3*), *Cryptochrome* (*Cry1, Cry2*), and *Rev-erb* (*Nr1d1, Nr1d2*) genes at the beginning of the feedback cycles. Protein dimer formed by PER and CRY suppresses the transcriptional activity of BMAL1::CLOCK, closing the first feedback loop. The second feedback mechanism is achieved by the nuclear receptor REV-ERBs to repress the transcription of the *Arntl* (*Bmal1*) gene (and to a lesser extent on *Clock* gene)[1].

While many peripheral organs have circadian clocks, the identities of rhythmic outputs are considerably divergent across tissues[2–6], contributing to organ-specific physiology and disorders associated with circadian misalignment[7]. However, the molecular mechanisms underlying heterogeneous circadian rhythms remain unclear. Tissue-specific chromatin occupancy of the core clock transcription factors BMAL1::CLOCK and REV-ERBα has been described, identifying co-occupancy of tissue-specific transcription factors[8–10]. In the context of these prior studies, it is of interest to apply genetic approaches to ascertain whether tissue-specific TFs influence clock TFs' loading onto chromatin or the other way around.

In multicellular organisms, cells from different tissues exhibit specialized gene expression profiles in part achieved by physically sequestering unnecessary genes into heterochromatin. Genes that are required for particular tasks of a cell type display accessible chromatin structure allowing for the binding of necessary machinery to facilitate gene expression[11]. Chromatin remodeling that opens condensed chromatin structures is initiated by the recruitment of lineage-specifying pioneer transcription factors to their target DNA sequences at enhancers. The pioneer TFs recruit histone methyltransferases MLL3/4 to deposit histone mark H3K4me1, whereby the condensed DNA wrapped around histones is loosened[12]. Completely activated enhancers feature bimodal distribution of histone modifications H3K4me1 and H3K27ac, nucleosome depletion, and recruitment of other transcription factors and coactivators[13]. Instead of being simply correlated with chromatin accessibility, H3K4me1 has an active regulatory role by serving as docking sites for chromatin remodelers[14]. Due to the activity of ATP-dependent chromatin remodelers[15] and three-dimensional chromatin folding[16], chromatin remodeling commonly creates extended accessibility beyond the central nucleosomes pioneer TFs bind.

With most (~16%) transcripts exhibiting circadian expression, the liver is the primary organ controlled by the circadian clock[4]. The hepatic circadian transcripts are highly organ-specific and involved in most principal functions of the liver, including glucose homeostasis, lipogenesis, bile acid synthesis, mitochondrial biogenesis, oxidative metabolism, amino acid turnover, and xenobiotic detoxification. Indeed, environmental or genetic disruption of the circadian clock exacerbates the development of liver diseases such as non-alcoholic fatty liver disease (NAFLD), hepatitis, cirrhosis, and hepatocellular carcinoma (HCC)[17]. The hepatocyte nuclear factor 4A (HNF4A) is a nuclear receptor specifically expressed in the liver, kidney, pancreas, and intestinal tracts[18]. Mutation or dysregulation of the *Hnf4a* gene is associated with human diseases such as maturity-onset diabetes of the young (MODY) and HCC[19,20]. Whole-body *Hnf4a* knockout resulted in embryonic lethality, and liver-specific knockout mice displayed severe hepatocyte differentiation defects and premature death by 8 weeks of age[21–23]. We previously demonstrated that HNF4A modulates peripheral circadian clocks in cell cultures[24].

Here, we further interrogate the interface between HNF4A and the circadian clock in the liver tissue where they both play critical roles. We find that HNF4A supervises BMAL1 chromatin binding seemingly by remodeling chromatin accessibility. Synchronized with HNF4A recruitment[24], mouse liver displayed increased genome-wide chromatin accessibility during the night. Furthermore, the circadian clock contributes to chromatin remodeling likely through regulating HNF4A. Our results reveal a collaborative effort between HNF4A and the clock machinery in shaping tissue-specific chromatin landscape and circadian rhythms that are vital for liver biology.

## Results

**BMAL1 chromatin binding is attenuated in the *Hnf4a* knockout liver.** Previously we discovered an extensive genome-wide colocalization of HNF4A and BMAL1::CLOCK in the mouse liver[24]. While physical interactions and genome-wide co-occupancy between the diurnal regulatory machinery and tissue-specific transcription factors have been reported[25–27], to our knowledge, how the tissue-specific factors may affect BMAL1::CLOCK recruitment has not been studied. To investigate the influence of HNF4A on BMAL1::CLOCK chromatin occupancy and circadian rhythms, we crossed *Hnf4a* floxed mice[22] with Albumin-Cre mice and Per2-luciferase mice in the same C57BL/6J background to generate liver-specific *Hnf4a* knockout (*Hnf4a*fl/fl *Alb-Cre*+/− *Per2-luc*+/+; HKO) and control (*Hnf4a*fl/fl *Alb-Cre*−/− *Per2-luc*+/+; Ctrl) mice (see the "Methods" section). In the HKO liver, RT-qPCR confirmed a ~75% decrease in *Hnf4a* transcript level accompanied by downregulation of the classic HNF4A target genes *ApoC3, Fabp1, Ppara*, and *Hnf1a* (Supplementary Fig. 1a). The liver-to-body-weight ratio was significantly increased for the HKO mice (Supplementary Fig. 1b). Histopathological analyses revealed extensive vacuolization in the HKO hepatocytes and marked lipid accumulation throughout the liver tissue (Supplementary Fig. 1c). Remarkably, in contrast with the premature lethality of HKO mice constructed with Albumin-Cre mice in the FVB genetic background[22], the HKO mice we constructed here live to at least the age of 9 months. The *Hnf4a* knockout liver exhibited more severe pathological lesions and greater changes in gene expression in male mice than the female[22,28], although the HCC development rate was sex-independent[29]. To eliminate sex as a confounder, we used male mice throughout the study. We mapped genome-wide BMAL1 binding profiles in liver samples collected from three HKO mice and three control mice at ZT6 when BMAL1 binding reaches maximum intensity[30]. Principal component analyses (PCA) of the three ChIP-seq replicates revealed clustering of samples from the same genotype (Supplementary Fig. 2a). Surprisingly, about 79% (5273 out of 6660) of the total BMAL1 peaks were prominently attenuated by *Hnf4a* removal, including ones located within the E-box-containing core clock genes (Fig. 1a, b). In addition to the clock genes, KEGG and gene ontology (GO) pathway analyses of the HKO-reduced BMAL1 binding genes (Supplementary Data 1) identified enrichment of metabolic pathways, such as glucose and cholesterol metabolism, especially when compared with the unchanged binding sites (Supplementary Fig. 2b, c). Therefore, circadian regulation of these key tissue-specific nodes[17] is supervised by HNF4A. The strong impact HNF4A exerted on BMAL1::CLOCK cistrome seemed to occur post-translationally, because BMAL1 transcript and protein levels were not reduced but rather moderately increased in the HKO liver, potentially related to downregulated *Nr1d1* and *Nr1d2* encoding transcriptional repressors of *Bmal1* (Fig. 1c, d).

Motif analysis of the HKO-reduced BMAL1 peaks indicated an enrichment of the HNF4A-binding motif, apart from the E-box

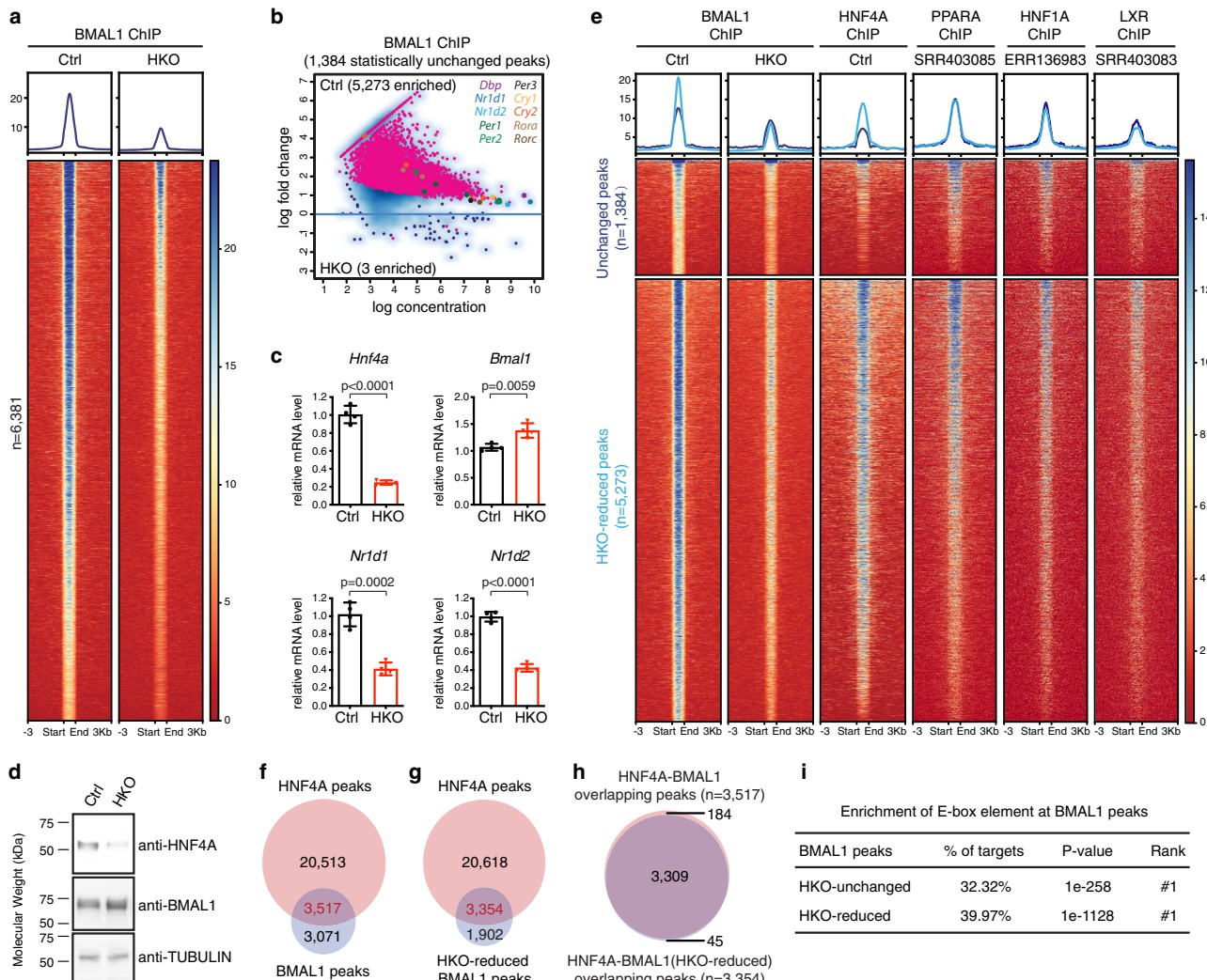

**Fig. 1 BMAL1 chromatin binding is attenuated in the *Hnf4a* knockout liver. a** Heatmap of BMAL1 ChIP-seq signals at ZT6 in control (left) or HKO (right) liver centered at all BMAL1 peaks in control liver. Peaks are ordered vertically by signal strength. **b** MA plot showing differential BMAL1 occupancy in control and HKO livers, using a threshold of FDR < 0.05. The *x*-axis represents the mean number of reads (log scaled) within the peaks across all samples. The *y*-axis represents the log fold change between the two samples. BMAL1 bindings at the core clock genes are highlighted. **c** Transcript level of genes was determined by RT-qPCR using liver samples isolated from control or HKO mice at ZT6. Displayed are the means ± SD (*n* = 4) normalized to *Rplp0* expression levels. Statistical significance was determined by a two-tailed Student's *t*-test. **d** Protein levels were determined by western blot analysis using liver samples isolated from control or HKO mice at ZT6. Two independent experiments were repeated with similar results. **e** BMAL1 peaks in control and HKO livers were partitioned into three categories with DiffBind (the HKO-enriched group has only three peaks and could not be plotted), and then the corresponding TF occupancy at each BMAL1 binding site was plotted. Each horizontal line represents a single BMAL1 binding site. Peaks were ordered vertically by the strength of the BMAL1 ChIP signal in the control liver. **f** Venn diagram showing the overlap between all BMAL1 binding sites (at ZT6) and all HNF4A binding sites (at ZT16). Overlapping peaks were identified using the mergePeaks command in HOMER (see the "Methods" section). Note that the peak numbers may not add up exactly since the function automatically resolves redundant overlaps by dropping one fragment during analysis. **g** Venn diagram showing the overlap between BMAL1-binding sites that were significantly reduced in HKO liver (at ZT6) and all HNF4A-binding sites (at ZT16). **h** Venn diagram showing the overlap between the HNF4A-BMAL1 co-occupancy sites identified in (**f**) and (**g**). **i** A summary of de novo motif analysis showing significance values of E-box enrichment at the HKO-unchanged or HKO-reduced BMAL1 peaks.

element (Supplementary Fig. 2d). We parsed all BMAL1 peaks into three groups based on signal variation in response to *Hnf4a* knockout: ones that were reduced (5273 peaks), enhanced (3 peaks), or not significantly changed (1384 peaks). On average, BMAL1 peaks of higher intensity tended to be more responsive to *Hnf4a* ablation (Fig. 1e). We also plotted HNF4A ChIP-seq signals when they reach maximum at ZT16[24] at each position of the BMAL1 peaks, finding HNF4A to display higher accumulation at the HKO-reduced BMAL1 peaks relative to the unchanged peaks (Fig. 1e). In contrast, for transcription factors PPARA, HNF1A, and LXR that were downregulated upon *Hnf4a* removal

(Supplementary Fig. 1a), by analyzing legacy ChIP-seq data[31,32], we did not observe their differential accumulation at BMAL1 peaks (Fig. 1e). Consistently, their binding motifs ranked far lower than the HNF4A-binding sequence at the HKO-reduced BMAL1 peaks (Supplementary Fig. 2d). The distance from a BMAL1 peak to the nearest HNF4A peak was significantly smaller in general for the HKO-reduced BMAL1 binding sites than unchanged ones (Supplementary Fig. 2e). Out of the 3517 BMAL1 peaks that colocalize with HNF4A occupancy, 3309 (94%) were greatly reduced by *Hnf4a* removal (Fig. 1f–h and Supplementary Fig. 2f). These data collectively indicate that

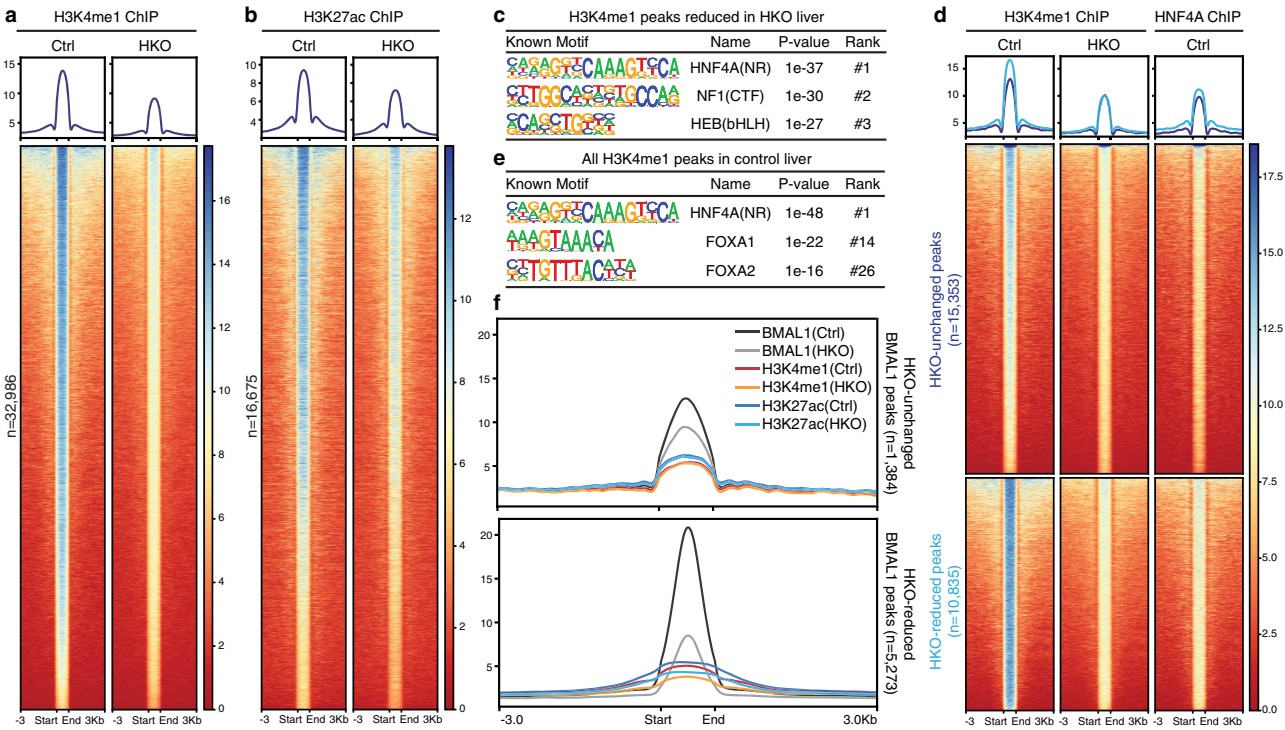

**Fig. 2 Hnf4a knockout alters the genome-wide epigenetic landscape. a**, **b** Heatmap of H3K4me1 (**a**) or H3K27ac (**b**) ChIP-seq signals at ZT6 in control (left) or HKO (right) liver centered at all peaks in control liver. Peaks are ordered vertically by signal strength. **c** Motif analysis of HKO-deprived H3K4me1 sites. Known consensus motifs are shown with corresponding enrichment significance values. **d** H3K4me1 peaks in control and HKO livers were partitioned into three categories with DiffBind (the HKO-enriched group has only 23 peaks and could not be plotted), and then the corresponding HNF4A occupancy (at ZT16) at each H3K4me1 site was plotted. Each horizontal line represents a single H3K4me1 site. Peaks were ordered vertically by the strength of the H3K4me1 ChIP signal in control liver. **e** Motif analysis of all H3K4me1 marked sites in the control liver. Known consensus motifs are shown with corresponding enrichment significance values. **f** Metaplot showing the average intensity of BMAL1, H3K4me1, and H3K27ac ChIP-seq signals (all at ZT6) in control or HKO livers surrounding HKO-unchanged (upper panel) or HKO-reduced (lower panel) BMAL1 peak centers.

HNF4A directly regulates global BMAL1 chromatin binding in the mouse liver. The underlying mechanisms do not involve gene expression regulation but are likely achieved on chromatin in a spatially restricted manner.

**Hnf4a knockout alters genome-wide epigenetic landscape.** The cooperative loading of transcription factors may involve two mechanisms: (1) a simultaneous loading mediated by protein–protein interactions; (2) a sequential loading that requires a pioneer TF to open up local chromatin for other factors to bind[13]. Notably, we detected physical interactions between BMAL1 and HNF4A in liver cells[24]. To evaluate the possibility of HNF4A recruiting BMAL1 to the genome, we compared enrichments of the E-box element at HKO-unchanged and HKO-reduced BMAL1 binding sites. The "% of targets" and "p-value of enrichment" reported by HOMER analysis indicated that the E-box sequence was present at a similar frequency within the two categories of BMAL1-binding sites (Fig. 1i). Moreover, there were a considerable fraction (1902/5256 = 36%) of HKO-reduced BMAL1-binding sites indeed not displaying exactly overlapping HNF4A occupancy (Fig. 1g). Therefore, it is unlikely for the HNF4A-BMAL1 physical interactions to be generally responsible for the HNF4A-dependent BMAL1 occupancy. We were prompted to ask if HNF4A acts as a pioneer TF and facilitates the accessibility of a broad range of chromatin that is a prerequisite for BMAL1 binding to occur.

The chromatin loading of a pioneer TF initiates increases in accessible/primed enhancers marked by H3K4me1 and subsequent chromatin activation marked by H3K27ac[13]. Therefore, the intensity of H3K4me1 and H3K27ac defines chromatin landscape and is indicative of pioneer TFs' activity. In agreement with our prediction, we observed a clear reduction in genome-wide H3K4me1 and H3K27ac deposition upon Hnf4a knockout (Fig. 2a, b, and Supplementary Fig. 3a, b), with the HNF4A-binding motif overrepresented at the HKO-reduced sites for both histone marks (Fig. 2c and Supplementary Fig. 3c). To interrogate to what extent HNF4A is involved in the early steps of chromatin remodeling, we looked into H3K4me1 and found it generally reduced at HNF4A-binding sites upon Hnf4a knockout (Supplementary Fig. 3d). In addition, HNF4A tended to accumulate more intensively at the H3K4me1 sites that would be significantly reduced by Hnf4a knockout (about 41.3% of total peaks), relative to the unchanged H3K4me1 sites (Fig. 2d). The distance from an H3K4me1 peak to the nearest HNF4A peak was noticeably smaller for the HKO-reduced H3K4me1 sites (Supplementary Fig. 3e). Similarly, the extent of H3K27ac loss in the HKO liver was positively correlated with the intensity of local HNF4A binding (Supplementary Fig. 3f). Motif analysis of all H3K4me1-marked regions in the control liver revealed maximal enrichment of the HNF4A-binding motif (Fig. 2e), in agreement with a global profiling finding HNF4A occupancy overrepresented in accessible regions of liver chromatin[33]. Taken together, HNF4A potentially serves as a key pioneer factor remodeling the active chromatin landscape in the liver. Of note, we found local deposition of H3K4me1 and H3K27ac marks was specifically reduced by Hnf4a knockout at the HKO-reduced BMAL1 sites (Fig. 2f and Supplementary Fig. 3g), supporting a working model that HNF4A supervises BMAL1 loading by helping establish a permissive chromatin landscape.

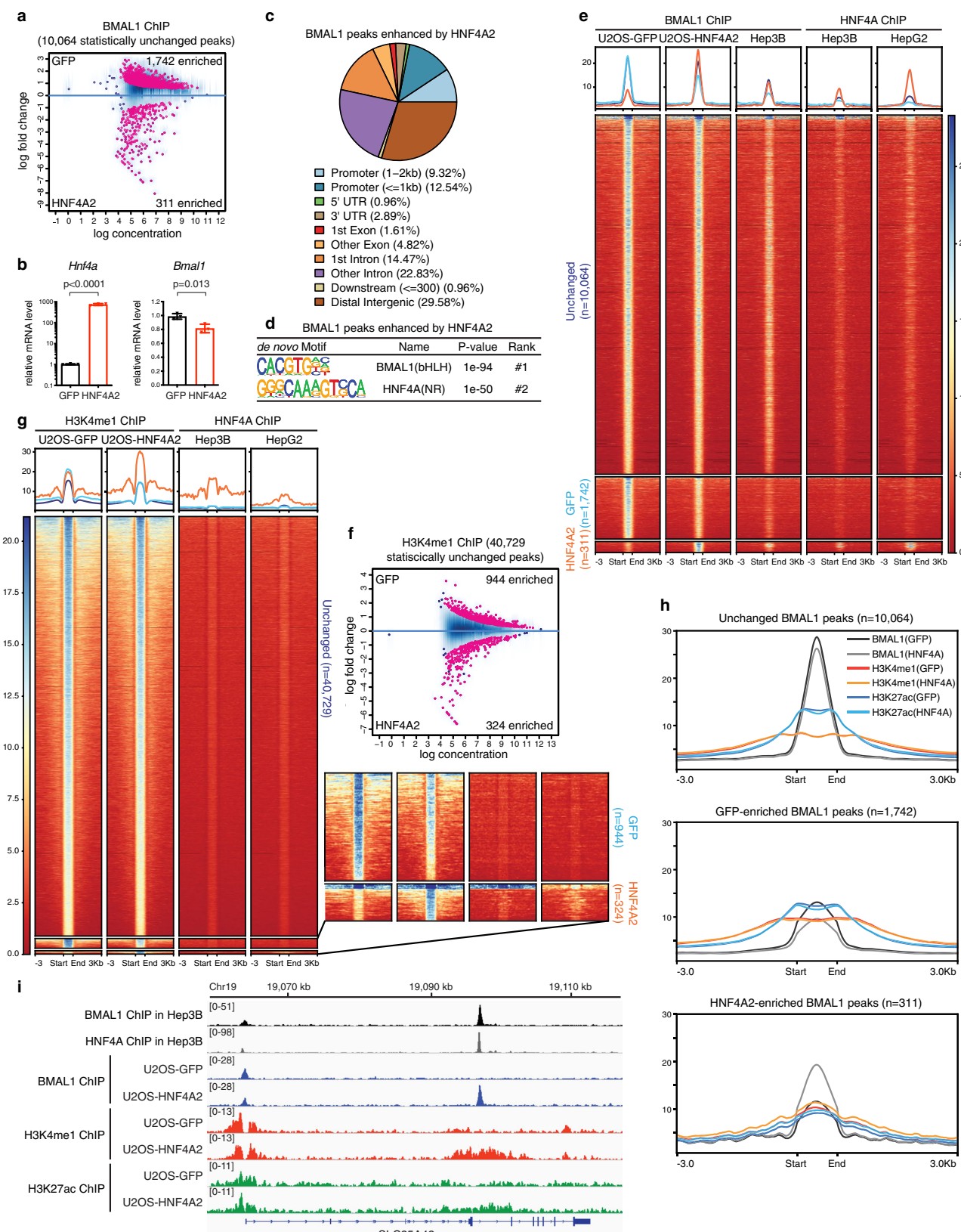

**Ectopic HNF4A expression reprograms epigenetic landscape and induces tissue-specific BMAL1 bindings**. Next, we sought to assess BMAL1 cistromes before and after HNF4A action in a biological system that has never been exposed to HNF4A protein. We ectopically expressed the adult isoform HNF4A2 in human bone osteosarcoma epithelial U2OS cells where the endogenous

*Hnf4a* expression is negligible[34]. 1742 BMAL1 peaks were moderately reduced by HNF4A2 expression (Fig. 3a and Supplementary Fig. 4a), seemingly resulting from downregulation of the *Bmal1* transcription (Fig. 3b). In the meanwhile, we identified 311 BMAL1-binding events that were significantly enhanced or gained de novo in response to HNF4A2 expression, compared

**Fig. 3 Ectopic HNF4A2 expression reprograms epigenetic landscape and induces tissue-specific BMAL1 bindings. a** MA plot showing differential BMAL1 occupancy in U2OS-GFP and U2OS-HNF4A2 cells, using a threshold of FDR < 0.05. The *x*-axis represents the mean number of reads (log scaled) within the peaks across all samples. The *y*-axis represents the log fold change between the two samples. **b** Transcript level of genes in U2OS-GFP or U2OS-HNF4A2 cells was determined by RT-qPCR. Displayed are the means ± SD (*n* = 3 cell culture wells) normalized to *Rplp0* expression levels. Statistical significance was determined by a two-tailed Student's *t*-test. **c** Distribution of genomic annotations of HNF4A2-enhanced BMAL1 peaks. **d** Motif analysis of HNF4A2-enhanced BMAL1 binding sites. de novo consensus motifs are shown with corresponding enrichment significance values. **e** BMAL1 peaks in U2OS-GFP and U2OS-HNF4A2 cells were partitioned into three categories with DiffBind. Then the corresponding BMAL1 and HNF4A occupancy in Hep3B or HepG2 cells were plotted by centering at each BMAL1-binding site in U2OS cells. Each horizontal line represents a single BMAL1-binding site in U2OS. Peaks were ordered vertically by the strength of the BMAL1 ChIP signal in U2OS. **f** MA plot showing differential H3K4me1 occupancy in U2OS-GFP and U2OS-HNF4A2 cells, using a threshold of FDR < 0.05. The *x*-axis represents the mean number of reads (log scaled) within the peaks across all samples. The *y*-axis represents the log fold change between the two samples. **g** H3K4me1 peaks in U2OS-GFP and U2OS-HNF4A2 were partitioned into three categories with DiffBind. Then the corresponding HNF4A occupancy in Hep3B or HepG2 cells was plotted by centering at each H3K4me1 site. Each horizontal line represents a single H3K4me1 site. Peaks were ordered vertically by the strength of H3K4me1 ChIP signal in U2OS. Heatmaps of GFP- and HNF4A2-enriched peaks are highlighted in the inset. **h** Metaplot showing average intensity of BMAL1, H3K4me1, or H3K27ac ChIP-seq signals in U2OS-GFP or U2OS-HNF4A2 cells surrounding centers of BMAL1 peaks of indicated groups. **i** IGV genome tracks showing BMAL1, HNF4A, H3K4me1, and H3K27ac enrichment at the *SLC25A42* gene in indicated cells, based on normalized ChIP-seq read coverage. Track heights are indicated.

with the GFP expression group (Fig. 3a and Supplementary Fig. 4a). These HNF4A2-induced BMAL1 peaks were more frequently located at distal or intronic enhancer regions (Fig. 3c) and enriched with the HNF4A-binding motif ranking second only to the E-box element (Fig. 3d). To interrogate the biological relevance of HNF4A2-induced BMAL1 bindings, we examined whether they occur in cells where HNF4A is naturally expressed. BMAL1 and HNF4A ChIP-seq signals from human liver cancer Hep3B or HepG2 cells were plotted correspondingly at each position of the BMAL1 binding sites we just profiled in U2OS-GFP and U2OS-HNF4A2 cells. Interestingly, BMAL1 ChIP signals in Hep3B cells displayed an analogous pattern to the U2OS-HNF4A2 dataset, i.e. signals at the U2OS-HNF4A2-enriched peak sites were stronger than those at the U2OS-GFP-enriched ones (Fig. 3e), indicating the U2OS-HNF4A2-induced BMAL1 peaks to be specifically expressed in liver cell cultures. Furthermore, endogenously expressed HNF4A in Hep3B or HepG2 cells was found to accumulate more abundantly at the U2OS-HNF4A2-induced BMAL1 peaks than the other sites (Fig. 3e). Therefore, the BMAL1-binding events we have induced in U2OS cells by introducing genome-wide occupancy of HNF4A2 may represent a true aspect of tissue-specific BMAL1 cistromes.

Chromatin landscape was confirmed to be remodeled by HNF4A2 expression, according to ChIP-seq profiling of H3K4me1 and H3K27ac (Fig. 3f and Supplementary Fig. 4b–d). In line with that observed at the gained BMAL1 peaks, the HNF4A2-enhanced H3K4me1 and H3K27ac sites were more likely located in distal or intronic enhancer regions (Supplementary Fig. 4e, f), concordant with a general recognition that lineage-specifying transcription factors exert physiologic effects through interactions with tissue-specific enhancers[13]. The U2OS-HNF4A2-enhanced H3K4me1 sites were confirmed to enrich more HNF4A occupation than the other sites in liver cells (Fig. 3g), suggesting that HNF4A2 binding is directly responsible for the induced H3K4me1 deposition. The subset of H3K4me1 sites that were mildly reduced by HNF4A2 expression, considering the minimal on-site HNF4A localization in liver cells (Fig. 3g), likely resulted from indirect effects of HNF4A2 ectopic expression. Lastly, distinct from the other BMAL1 peaks, the HNF4A2-induced BMAL1 peaks were marked by locally enhanced deposition of H3K4me1 and H3K27ac upon HNF4A expression (Fig. 3h and Supplementary Fig. 4g). To exhibit the HNF4A2-reprogrammed BMAL1, H3K4me1, and H3K27ac peaks in higher resolution, we present genome tracks of representative genes (*SLC25A42*, *DOK4*, *CDHR2*, and *PLPP3*) in Fig. 3i and Supplementary Fig. 5. The fetal HNF4A isoforms lacking the N-terminal activation domain AF-1 relative to the

adult isoforms are specifically expressed in the embryonic liver and diseased liver. They occupy much the same set of genome loci as the adult isoforms do yet exhibit a lower transcriptional activity[35,36]. We found that ectopically expressing the fetal isoform HNF4A8 induced tissue-specific BMAL1 binding likewise (Supplementary Fig. 6), arguing that HNF4A-regulated BMAL1 recruitment is invariable during liver development and disease transition. Taken together, we programmed tissue-specific BMAL1 bindings by remodeling E-box-containing enhancers which are otherwise actively masked by nucleosomes. Existing literature has demonstrated that functional BMAL1::CLOCK occupancy at circadian enhancers closely correlates with the oscillation of the target genes[37,38]. We speculate that in some cases HNF4A expression alone is not enough for achieving efficient chromatin opening and the presence of additional chromatin remodeling factors is necessary.

## Circadian rhythms are disturbed by *Hnf4a* knockout and HNF4A-MODY mutation.

We previously showed that *Hnf4a* knockdown caused varying degrees of circadian rhythm disruption in cell cultures including period shortening and complete arrhythmicity[24]. Consistently, tissue explants of HKO liver exhibited a shorter period of *Per2-Luc* oscillation ex vivo (Fig. 4a, b). Control and HKO liver tissues were collected every four hours from mice housed under a 12-h light:12-h dark cycle (LD 12:12). RT-qPCR quantification of the core clock transcripts revealed robust circadian oscillations in the control liver, while a dampening was observed after *Hnf4a* ablation (Fig. 4c). This phenotype was especially clear for *Dbp*, *Nr1d1*, and *Nr1d2* (Fig. 4c), genes that are distinct from the other E-box-containing clock genes and lose expression in the *Bmal1* knockout mice[37,39]. Downregulation of the three BMAL1::CLOCK-dependent genes was confirmed by dimmed local H3K4me1 and H3K27ac signals and associated with dysregulated BMAL1 recruitment (Fig. 4d and Supplementary Fig. 7). Since *Hnf4a* is minimally expressed outside the liver, kidney, pancreas, and intestinal tracts, we do not expect it to act on the master circadian clock in the SCN or animal behaviors.

*Hnf4a* mutations were frequently identified in patients with MODY, a rare form of diabetes[19]. In agreement, disrupting *Hnf4a* expression in mouse islets or insulinoma cells resulted in impaired glucose-stimulated insulin secretion[40]. Interestingly, insulin secretion by pancreatic β cells is rhythmic, and perturbation of the circadian cycles contributes to diabetes[8,41]. Our results provide an excellent opportunity for investigating whether HNF4A-MODY mutations connect clock dysregulation

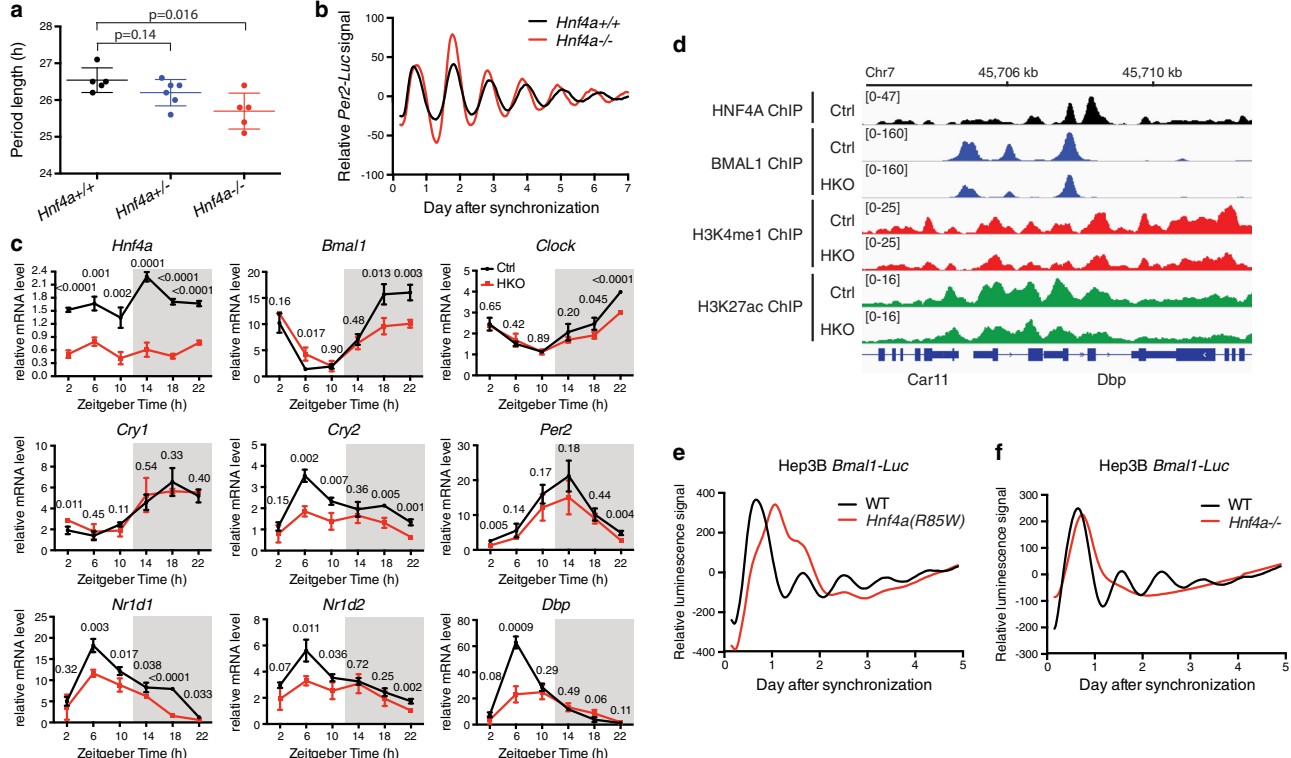

**Fig. 4 Circadian rhythms are disturbed by *Hnf4a* knockout and HNF4A-MODY mutation. a, b** Liver tissue explants were isolated from mice of indicated genotypes and recorded for *Per2-Luc* bioluminescence. Period lengths of *Per2-Luc* oscillation were plotted (means ± SD, *n* = 5 or 6) and statistical significance was determined by two-tailed Student's *t*-test (**a**). Representative bioluminescence records show *Per2-Luc* circadian profiles in control or HKO liver (**b**). **c** Control and HKO mouse livers were harvested at 4-h intervals over the course of 24 h. Transcript level of genes was analyzed by using RT-qPCR. Displayed are the means ± SD (*n* = 3 or 4) normalized to non-oscillating *Rplp0* expression levels. *P*-values determined by two-tailed Student's *t*-test were displayed. **d** IGV genome tracks showing HNF4A (at ZT16), BMAL1 (at ZT6), H3K4me1 (at ZT6), and H3K27ac (at ZT6) enrichment at the *Dbp* gene in liver tissues, based on normalized ChIP-seq read coverage. Track heights are indicated. **e, f** Representative effect of *Hnf4a(R85W)* point mutation (**e**) or *Hnf4a* knockout (**f**) on *Bmal1-Luc* oscillation in human Hep3B cells (*n* = 3).

to the development of diabetes. R85W is a mutation within the DNA-binding domain of HNF4A that was repeatedly identified in MODY patients[42,43]. To investigate this connection, we generated homozygous R85W mutation using CRISPR-Cas9 and surprisingly found the mutant cells to exhibit fundamentally disrupted circadian rhythms (Fig. 4e and Supplementary Fig. 8a), resembling cells carrying *Hnf4a* homozygous knockout (Fig. 4f and Supplementary Fig. 8b). Therefore, HNF4A-MODY patients may express dysregulated circadian rhythms which potentially contribute to the disease's pathogenesis and progression.

**HNF4A governs liver-specific circadian transcription.** The chromatin remodeling activity and rhythmic recruitment[24] of HNF4A prompted us to test whether the hepatic chromatin landscape is dynamically shaped throughout the day. We performed ChIP-seq analyses of H3K4me1 and H3K27ac with wild-type mouse livers collected at ZT16, the peak time of HNF4A binding[24], or the antiphase ZT6. Interestingly, the genome-wide deposition of H3K4me1 or H3K27ac was overall higher at ZT16 (Fig. 5a, b and Supplementary Fig. 9a, b). ATAC-seq that assesses genome-wide chromatin accessibility by probing open chromatin showed an analogous pattern (Fig. 5c). Indeed, our results indicating that chromatin accessibility in the liver is greater at night are in agreement with observations that the phases of cycling transcripts remarkably clustered between midnight and dawn in the developmentally related liver and kidney where HNF4A is tissue-specifically expressed[4,30]. Therefore, genome-wide HNF4A

occupancy, chromatin opening, and circadian output gene expression are in phase and potentially causally linked. Indeed, we identified the HNF4A-binding motif most enriched at ZT16-enhanced H3K4me1 or H3K27ac sites (Supplementary Fig. 9c–f). The night-time enhanced HNF4A recruitment potentially induces bursts of genome-wide gene expression by facilitating DNA accessibility by transcriptional machinery.

We identified 6,995 H3K4me1 peaks that were prominently stronger at ZT16 than ZT6 and 44,765 statistically unchanged peaks (Supplementary Fig. 9c). Relative to the unchanged peaks, genes with ZT16-enhanced H3K4me1 peaks were more likely involved in the circadian rhythm pathway, along with critical aspects of hepatic functions, in particular cholesterol metabolism, gluconeogenesis, insulin resistance, drug metabolism, and autophagy (Fig. 5d). Indeed, all of these cellular processes were characterized to operate under circadian control[17] and feature rhythmic expression of key regulatory genes[2]. Other than glucose metabolism, HNF4A is well characterized in the regulation of lipid and xenobiotic metabolisms[44,45]. HNF4A-MODY patients also exhibit liver disorders such as increased LDL cholesterol levels owing to altered expression of apolipoprotein genes[46,47]. Therefore, central mechanisms underlying HNF4A-regulated hepatic metabolisms may involve circadian regulation whereby HNF4A synchronizes metabolic activities with active food intake after dark.

To further interrogate HNF4A roles in tissue-specific circadian transcription, we assessed circadian rhythms of genes that were significantly downregulated by *Hnf4a* knockout[48] or most

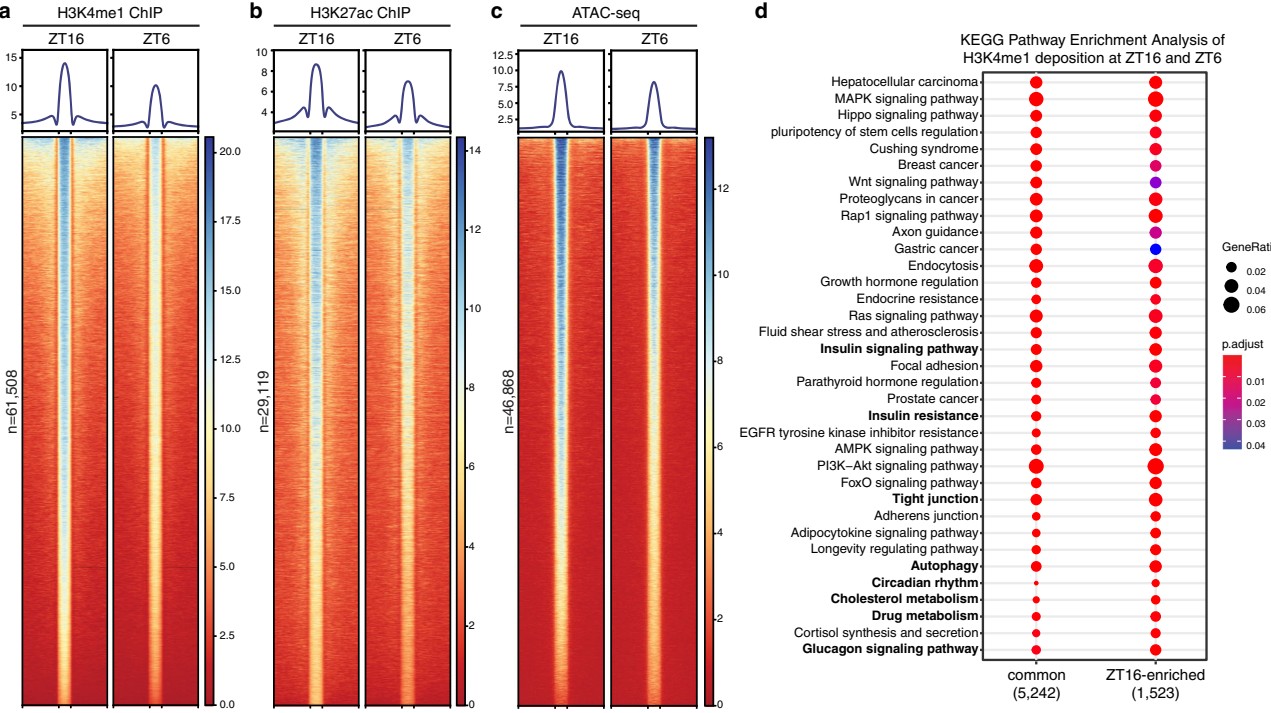

**Fig. 5 Mouse liver chromatin is more accessible at night, synchronized with HNF4A recruitment. a–c** Heatmap of H3K4me1 ChIP-seq (**a**), H3K27ac ChIP-seq (**b**), or ATAC-seq (**c**) signals within liver tissues sampled at ZT16 (left) or ZT6 (right) and centered at all peaks of ZT16. Peaks are ordered vertically by signal strength. **d** KEGG pathway analysis was performed for genes having ZT16-ZT6-common or ZT16-enriched H3K4me1 peaks as defined in Supplementary Fig. 9c.

differentially expressed at all time points by *Bmal1* knockout[49]. CircaDB[50] identified 38% of HNF4A-downregulated and 37% of BMAL1-regulated genes robustly rhythmic (Supplementary Data 2), higher than the ratio of 16% for general hepatic transcripts[4]. The HNF4A-regulated circadian transcripts tend to peak at the pre-dawn "rush hours" (Supplementary Fig. 10a, b) and are highly enriched in pathways of circadian rhythm, lipid and cholesterol metabolism, amino acid metabolism, redox reactions, and liver development (Supplementary Fig. 10c–f), strongly arguing that HNF4A regulates tissue-specific circadian rhythms.

**The circadian clock modulates genome-wide DNA binding of HNF4A.** To evaluate how HNF4A activity is supervised by the circadian clock, we first induced chronic circadian disruption in mice by performing a jet lag protocol for 4 weeks. At the end of the treatment, remarkably, night-time enhanced HNF4A recruitment was reversed, i.e. HNF4A ChIP-seq signals at ZT16 were no longer greater than those at ZT4[24] (Fig. 6a and Supplementary Fig. 11a). Therefore, the daily cycle of HNF4A chromatin loading is generated by the circadian clock. We then assessed whether BMAL1, in turn, regulates HNF4A chromatin binding by using the liver-specific *Bmal1* knockout mouse model[51] (see the "Methods" section). We mapped genome-wide DNA binding of HNF4A at ZT16 in liver samples collected from liver-specific *Bmal1* knockout (*Bmal1*^*fl/fl* *Alb-Cre*^+/−; BKO) or control mice (*Bmal1*^*fl/fl* *Alb-Cre*^−/−; Ctrl), identifying about 14% (4576 out of 32,201) of total HNF4A ChIP-seq peaks reduced and about 1% (321 out of 32,201) enhanced in BKO liver (Fig. 6b, c and Supplementary Fig. 11b). KEGG and GO term analyses of BKO-reduced HNF4A-binding sites (Supplementary Data 3) revealed genes involved in cancer pathogenesis among most enriched. Other overrepresented categories included Wnt/β-catenin signaling and cell cycle pathways (Supplementary

Fig. 11c, d). HNF4A inhibits Wnt/β-catenin signaling and cell cycle progression, potentially underlying its tumor-suppressive roles[52]. In comparison, genome-wide binding of the well-characterized hepatic pioneer factor FOXA2[13] was barely affected by *Bmal1* knockout (Supplementary Fig. 11e, f). BMAL1 co-occupancy was only slightly more enriched at the BKO-reduced HNF4A binding sites (19.2% colocalized with BMAL1 binding) than the control sites (14.0% for total HNF4A peaks; 10.3% for BKO-unchanged peaks) (Supplementary Fig. 11g, h), therefore, it is unlikely for chromatin recruitment mediated by protein–protein interactions to play a dominant role in the regulation. Instead, we found *Hnf4a* transcription steadily down-regulated by 20–30% upon *Bmal1* removal at all sampling times (Fig. 6d). Since BMAL1 directly binds to the *Hnf4a* gene body (Supplementary Fig. 11i), BMAL1::CLOCK likely modulates HNF4A chromatin binding through transcriptional regulation. Analogous to *Per2* transcripts, although dampened, *Hnf4a* oscillation was not abolished by *Bmal1* removal (Fig. 6d). Since night-enhanced *Hnf4a* expression was not altered by fasting[24], mechanisms rather than feeding behavior may be involved in clock-independent *Hnf4a* oscillation.

**Bmal1 knockout alters epigenetic landscape seemingly due to attenuated HNF4A activity.** To assess whether chromatin remodeling is responsible for BMAL1-regulated HNF4A genome binding, we profiled genome-wide locations of H3K4me1 and H3K27ac at ZT16 in control or BKO liver tissues. Overall, the histone marks were not greatly changed by BKO (Fig. 7a, b and Supplementary Fig. 12a, b) especially when compared with HKO (Fig. 2a, b). Statistical analysis identified small subgroups that were significantly reduced or enhanced by *Bmal1* knockout (Fig. 7c, d). For instance, genes exhibiting significantly reduced histone modifications included *Nr1d2*; genes exhibiting

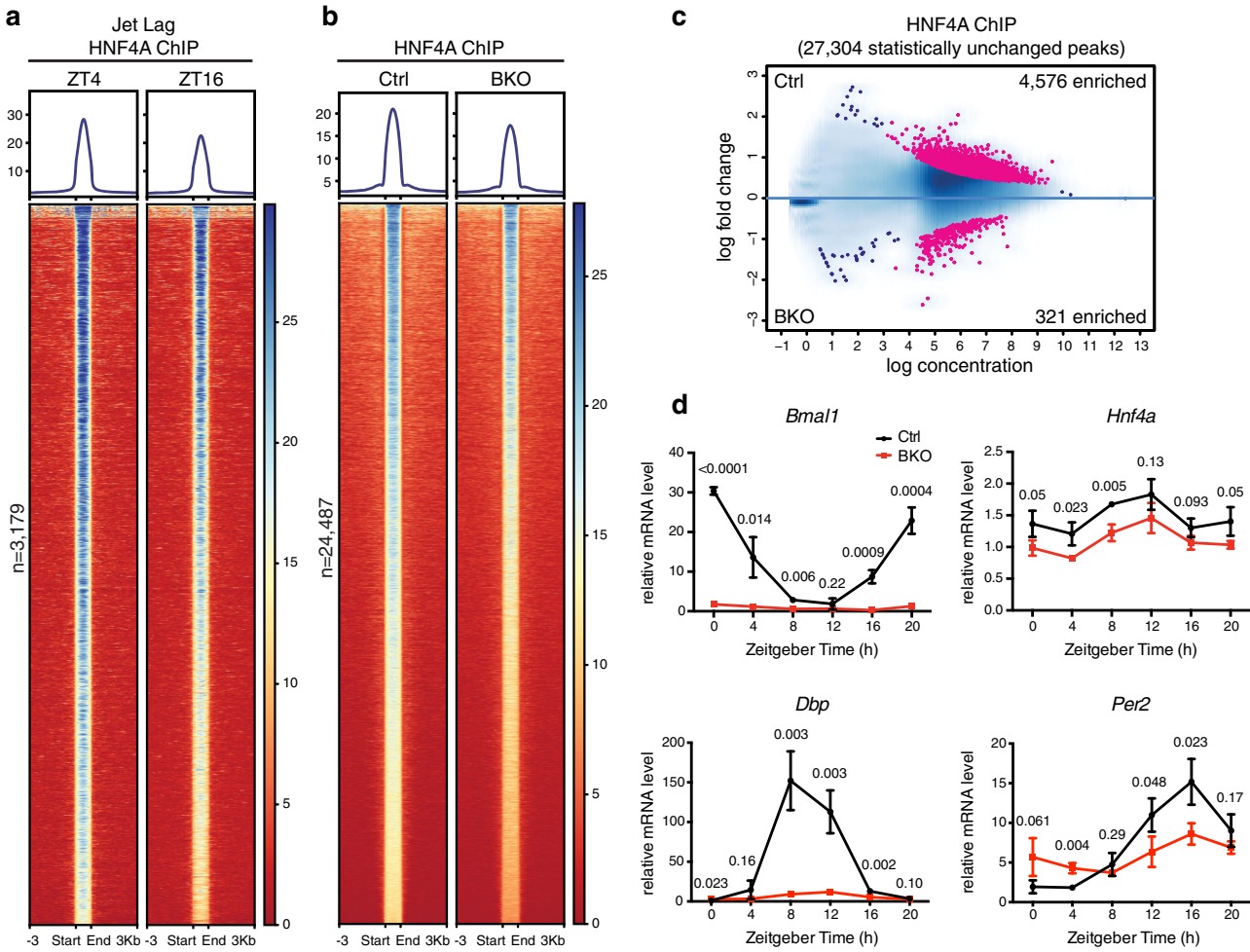

**Fig. 6 The circadian clock modulates genome-wide DNA binding of HNF4A. a** Heatmap of HNF4A ChIP-seq signals within liver tissues sampled at ZT4 (left) or ZT16 (right) after jet lag treatment and centered at all HNF4A peaks of ZT4. Peaks are ordered vertically by signal strength. **b** Heatmap of HNF4A ChIP-seq signals at ZT16 in control (left) or BKO (right) liver centered at all HNF4A peaks in control liver. Peaks are ordered vertically by signal strength. **c** MA plot showing differential HNF4A peaks in control and BKO livers, using a threshold of FDR < 0.05. The *x*-axis represents the mean number of reads (log scaled) within the peaks across all samples. The *y*-axis represents the log fold change between the two samples. **d** Control and BKO mouse livers were harvested at 4-h intervals over the course of 24 h. Transcript level of genes was analyzed by using RT-qPCR. Displayed are the means ± SD (*n* = 3) normalized to non-oscillating *Rplp0* expression levels. *P*-values determined by two-tailed Student's *t*-test were displayed.

significantly enhanced histone modifications included *Npas2* (Fig. 7e).

Motif analysis of the BKO-enhanced H3K4me1 (*n* = 264) or H3K27ac (*n* = 134) peaks identified the ROR response element (RORE), binding motif of transcriptional repressors REV-ERBs (Supplementary Fig. 12c, d). Interestingly, at the BKO-reduced H3K4me1 (*n* = 987) or H3K27ac (*n* = 101) peaks, we did not identify the E-box element but instead found an enrichment of nuclear receptor binding sites, with the HNF4A-binding motif top-ranked (Supplementary Fig. 12e, f). About 4.6% of total H3K4me1 peaks display colocalization with BMAL1 binding within a distance of 500 bp. This degree of BMAL1 colocalization remained similar for the BKO-unchanged (4.5%) and BKO-reduced (4.0%) subgroups of H3K4me1 sites (Fig. 7f and Supplementary Fig. 12g), indicating BMAL1 occupancy not enriched at the BKO-reduced H3K4me1 sites. In contrast, H3K4me1 peaks having HNF4A colocalization increased from a background of 19.0 to 27.5% for the BKO-reduced sites, and decreased to 16.6% for the BKO-unchanged sites (Fig. 7f and Supplementary Fig. 12h). We consider HNF4A co-occupancy enriched at the BKO-reduced H3K4me1 sites, given that only 42% of the HKO-reduced H3K4me1 peaks exhibited HNF4A

colocalization within the same distance of 500 bp (Supplementary Fig. 12i). We noticed that HNF4A occupancy was selectively reduced at the BKO-reduced H3K4me1 sites compared with the unchanged sites (Supplementary Fig. 12j). Importantly, we plotted H3K4me1 and H3K27ac ChIP-seq signals at each position of the BMAL1 or HNF4A peaks to find both histone marks specifically attenuated by BKO at HNF4A peaks (Fig. 7g) rather than at BMAL1 peaks (Fig. 7h). Therefore, BMAL1::CLOCK occupancy does not directly regulate active epigenetic modifications at ZT16 but through positively modulating HNF4A. Taken together, it is unlikely for BMAL1 to regulate HNF4A cistrome through chromatin remodeling.

## Discussion

Our findings demonstrate that HNF4A may act as a pioneer TF creating tissue-specific repertoires of accessible *cis*-regulatory elements. Consistently, the HNF4-binding element was top-scoring in accessible chromatin regions in the intestinal duodenal epithelium[53,54]. HNF4A was essential for maintaining active histone signature H3K27ac in the intestine and liver[53,55]. While HNF4A was an established fundamental liver development

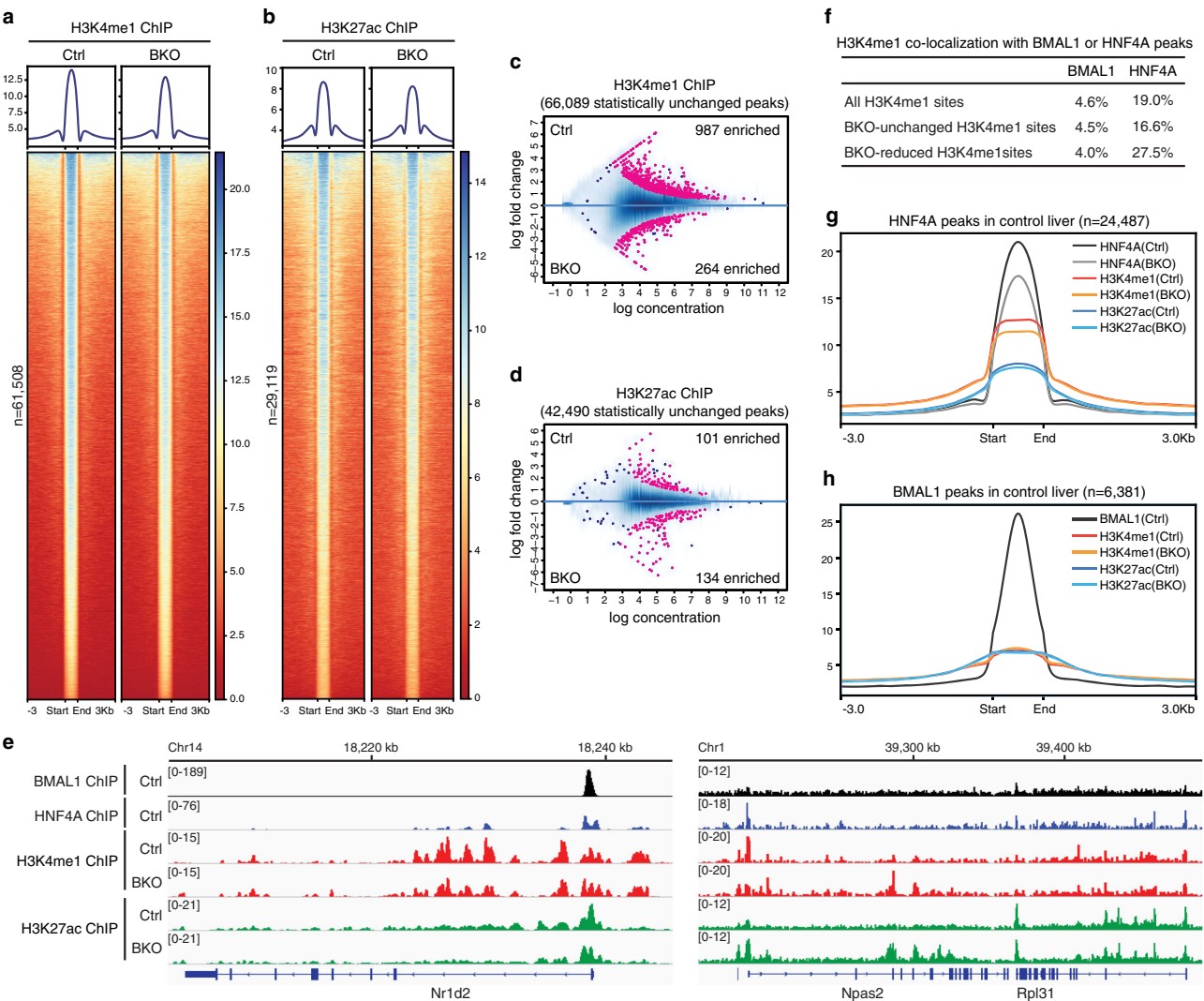

**Fig. 7 Bmal1 knockout alters the epigenetic landscape in the liver, seemingly due to attenuated HNF4A activity. a, b** Heatmap of H3K4me1 (**a**) or H3K27ac (**b**) ChIP-seq signals at ZT16 in control (left) or BKO (right) liver and centered at all peaks in control liver. Peaks are ordered vertically by signal strength. **c, d** MA plot showing differential H3K4me1 (**c**) or H3K27ac (**d**) peaks in control and BKO livers, using a threshold of FDR < 0.05. The x-axis represents the mean number of reads (log scaled) within the peaks across all samples. The y-axis represents the log fold change between the two samples. **e** IGV genome tracks showing BMAL1 (at ZT6), HNF4A (at ZT16), H3K4me1 (at ZT16), and H3K27ac (at ZT16) enrichment at *Nr1d2* and *Npas2* genes in control or BKO liver, based on normalized ChIP-seq read coverage. Track heights are indicated. **f** Percentages of three groups of H3K4me1 sites colocalizing with BMAL1 or HNF4A peaks. Peak numbers for percentage calculation are in Supplementary Fig. 12g, h. **g** Metaplot showing average intensity of HNF4A, H3K4me1, and H3K27ac ChIP-seq signals (all at ZT16) in control or BKO livers surrounding HNF4A peak centers in control liver. **h** Metaplot showing the average intensity of BMAL1 (at ZT6), H3K4me1 (at ZT16), and H3K27ac (at ZT16) ChIP-seq signals in control or BKO livers surrounding BMAL1 peak centers in control liver.

regulator, it was not as well characterized in the process of chromatin remodeling as another hepatic TF FOXA/ HNF3[13,56,57]. Nevertheless, we found the HNF4A-binding motif more enriched than the FOXA motifs in H3K4me1-positive liver genome regions (Fig. 2e). HNF4A was essential and to some extent sufficient for establishing liver-specific chromatin landscape (Figs. 2 and 3). Notably, among all hepatic TFs, HNF4A was the most important in converting human fibroblasts to hepatocyte-like cells (hiHeps)[58,59]. These observations collectively suggest that HNF4A remodels the chromatin landscape for active gene expression changes during development. To gain mechanistic insights, direct nucleosome binding studies will be needed in future to clarify whether HNF4A can independently displace histones like FOXA/HNF3 does or engages ATP-dependent enzymes to expand the "openness" of local chromatin[13].

The REV-ERB regulation of metabolic genes was reported to require chromatin recruitment by hepatic transcription factors[10,60]. The activity of pancreatic cycling gene expression displayed a correlation with the binding of the pancreas-specific transcription factor PDX1[8]. Despite these insights into the role of the lineage-specifying TFs, there has been a gap in understanding the molecular basis of tissue-specific rhythmicity whose misalignment is closely associated with organ-specific disorders[7]. By using loss-of-function and gain-of-function genetic models, we demonstrate that the lineage-specifying HNF4A is critical and in some cases sufficient for establishing liver-specific BMAL1 cistrome, seemingly independent of direct recruitment but by means of providing permissive chromatin structure. Our results may provide a molecular basis for tissue-specific BMAL1::CLOCK cistromes depending on the chromatin structures likely arising

from early events in tissue development. Systematic profiling of 20 diverse human cell types identified ~25% of genes displaying cell-type-specific expression that is explained by alterations in chromatin structures[61]. Our discoveries suggest that tissue-specific chromatin landscape profoundly shapes the circadian rhythms, providing a unifying mechanism for circadian rhythm heterogeneity across tissue types. We recently reported that the genome-wide BMAL1::CLOCK occupancy in glioblastoma stem cells was more expanded as compared with normal neural stem cells[62]. Given that chromatin structure alterations are prevalent in tumor tissues[63], our findings may provide additional insights into reprogrammed circadian clocks now found in cancer and many other disease states.

Liver-specific *Hnf4a* removal undermined BMAL1 occupancy at most of its target genes including the E-box-containing core clock genes (Fig. 1b), downregulated transcription of the BMAL1::CLOCK-dependent core clock genes *Dbp*, *Nr1d1*, and *Nr1d2* (Fig. 4c), and shortened the period of *Per2-Luc* oscillation (Fig. 4a, b). Potentially resulting from altered *Nr1d1* and *Nr1d2* expression, *Bmal1* transcription was upregulated in HKO liver (at ZT6) (Fig. 1c) and downregulated by HNF4A ectopic expression (Fig. 3b and Supplementary Fig. 6a). The adult HNF4A was reported to repress *Bmal1* expression less than the fetal form[64], potentially underlying the mild *Bmal1* upregulation in the HKO liver where the adult HNF4A is specifically targeted due to spatiotemporal expression of Albumin-Cre. It seems that the HNF4A actions on BMAL1::CLOCK activity are multilayered, including promoting the chromatin binding, transrepressing the transcriptional activity[24], and negatively regulating *Bmal1* transcription. These seemingly contradictory modes of action are indeed prevalently employed by circadian clock regulators so as to maintain circadian homeostasis, by virtue of the nature of the interlocking negative feedback loops[65]. For instance, CRY stabilization lead to suppression of BMAL1::CLOCK transcriptional activity and a simultaneous increase in *Bmal1* transcription which was largely attributable to downregulated *Rev-erb* genes[66]. Given that *Dbp*, *Nr1d1*, and *Nr1d2* were downregulated in the HKO liver (Fig. 4c), the chromatin remodeling activity of HNF4A seems to play a dominant role here by positively impacting BMAL1::CLOCK activities. The robust BMAL1::CLOCK transrepression activity we have characterized[24] can serve as a second mechanism for HNF4A to fine-tune circadian rhythms only after BMAL1::CLOCK is efficiently recruited to the target genes. In aggregate, our results indicate that HNF4A is a key modulator of the core circadian clock machinery.

Largely in agreement with prior studies[30,38] (Supplementary Fig. 9g–h), genome-wide H3K4me1 and H3K27ac deposition, as well as chromatin accessibility assessed by ATAC-seq, were increased during the night (Fig. 5a–c). The rhythmic recruitment of HNF4A may stimulate a synchronized day–night transition of chromatin accessibility, which intriguingly coincides with the predawn "rush hours" of circadian gene transcription in the liver[4,30]. Zhang et al.[4] profiled circadian transcriptomes of 12 different mouse organs and found the phase distribution of circadian transcripts in the liver and kidney to exhibit patterns distinct from the other 10 organs, i.e. being clustered between midnight and dawn. Given that out of the 12 organs investigated, only the liver and kidney indeed express the HNF4A protein, HNF4A is likely responsible for the unique repertoire and phase distribution of circadian outputs in the two organs. Even though the pioneer activity of HNF4A is higher at night, BMAL1, H3K4me1, and H3K27ac ChIP-seq signals were all considerably reduced at noon (ZT6) by *Hnf4a* knockout (Figs. 1 and 2). Therefore, HNF4A seems to determine the hepatic chromatin landscape from morning till night. Collectively, the chromatin remodeling activities of HNF4A may control tissue-specific circadian rhythms through two mechanisms: (1) to facilitate BMAL1::CLOCK recruitment during the day and secure the operation of the core clock; (2) to open chromatin maximally during the night and promote predawn-clustered expression of tissue-specific circadian outputs.

Finally, the rhythmic HNF4A genome binding was disrupted by chronic jet lag (Fig. 6a). BMAL1 promoted efficient genome binding of HNF4A, likely independent of protein–protein interactions or chromatin remodeling but through activating *Hnf4a* transcription (Figs. 6 and 7). *Bmal1* knockout only slightly altered H3K4me1 and H3K27ac at ZT16 (Fig. 7a–d). The RORE element was enriched at BKO-enhanced modification sites; the BKO-reduced modification sites did not enrich E-box element or BMAL1 colocalization but were associated with HNF4A binding (Fig. 7f–h). Therefore, BMAL1::CLOCK modulates hepatic epigenetic landscape potentially by activating target genes, namely *Rev-erbs* and *Hnf4a*. These results incidentally support our main finding that HNF4A shapes the hepatic chromatin landscape. The circadian clock regulates HNF4A transcription and rhythmic DNA binding whereby it contributes to the hepatic epigenetic landscape.

## Methods

**Animal experiments.** All animal care and experiments were performed under the institutional protocols approved by the Institutional Animal Care and Use Committee (IACUC, #20826) at the University of Southern California. *Hnf4a* floxed mice (The Jackson Laboratory #004665)[22] were crossed with Albumin-Cre mice (The Jackson Laboratory #003574) and Per2-luciferase reporter mice (The Jackson Laboratory #006852)[67] to obtain *Hnf4a*$^{fl/+}$;*Alb-Cre*$^{+/-}$;*Per2-luc*$^{+/+}$; and *Hnf4a*$^{fl/fl}$; *Alb-Cre*$^{-/-}$;*Per2-luc*$^{+/+}$ mice, which were then mated to obtain *Hnf4a*$^{fl/fl}$; *Alb-Cre*$^{+/-}$;*Per2-luc*$^{+/+}$ (HKO) and *Hnf4a*$^{fl/fl}$;*Alb-Cre*$^{-/-}$;*Per2-luc*$^{+/+}$ (Control) littermates. *Bmal1* floxed mice (The Jackson Laboratory #007668)[51] were crossed with mice expressing Albumin-Cre (The Jackson Laboratory #003574) to obtain *Arntl*$^{fl/+}$;*Alb-Cre*$^{+/-}$ and *Arntl*$^{fl/fl}$;*Alb-Cre*$^{-/-}$ mice which were then mated to obtain *Arntl*$^{fl/fl}$;*Alb-Cre*$^{+/-}$ (BKO) and *Arntl*$^{fl/fl}$;*Alb-Cre*$^{-/-}$ (Control) littermates. In all experiments, male mice between 10 and 12 weeks of age were used. In all experiments except the jet lag treatment, mice were housed in a room with controlled temperature of 21–23 °C and humidity of 35–40% under a 12-h light/12-h dark (LD) cycle with free access to food and water. The chronic jet lag treatment was performed by housing experimental mice in the light-tight circadian cabinet and switching lighting conditions between two light onset schedules which are apart by 8 h every 3 days from 7 to 11 week of age.

**Plasmid constructs, lentivirus production, and generation of stable cell lines.** *EGFP*, *Hnf4a2* (OriGene, #RC217863), and *Hnf4a8* (OriGene, #RC238302) genes were subcloned into the lentiviral vector pLV-EF1a-IRES-Ametrine. The constructed vectors were then co-transfected with the envelope and packaging plasmids into HEK 293T cells for lentivirus production. Viral supernatants were collected twice, at 48 and 72 h after transfection, pooled and filtered through 0.45 μm filters, and then added to U2OS cells for transduction. The positively transduced U2OS cells were selected by FACS sorting according to the Ametrine signals.

**Cell culture and circadian assays.** *Hnf4a* knockout and MODY *Hnf4a(R85W)* mutation in Hep3B cells expressing *Bmal1-Luc* reporter were generated using CRISPR-Cas9. HEK 293T, U2OS, and Hep3B cells were grown in complete DMEM (Life Technologies cat. #11995065) supplemented with 10% FBS and 1% penicillin/streptomycin. HepG2 cells were grown in Ham's F12 (Corning Cellgro 10-080-CV) supplemented with 10% FBS, and 1% penicillin and streptomycin. All cells were grown in a 37 °C incubator maintained at 5% $CO_2$. The Hep3B cell line was a gift from Dr. Michael Karin at UCSD, originally purchased from ATCC (HB-8064). HEK 293T, U2OS, and HepG2 cell lines were directly purchased from ATCC (CRL-3216, HTB-96, HB-8065).

For circadian assays, Hep3B cells were plated on 35-mm dishes and synchronized by a dexamethasone shock as previously described[24,62]. In brief, cell culture media was replaced with HEPES-buffered phenol-free DMEM media containing 100 nM dexamethasone and 100 μM D-luciferin. Dishes were covered with 40 mm glass coverslips (Fisher Scientific) and sealed with vacuum grease to prevent evaporation. Luminescence signals were monitored every 10 min using the LumiCycle luminometer (Actimetrics) at 37 °C without supplementary $CO_2$. Results shown are representative of at least three independent experiments.

**Liver explant circadian assays.** Mice were anesthetized with isoflurane and euthanized by cervical dislocation, and then livers were rapidly removed and kept

on ice. Liver sections (2–3 mm*2–3 mm) were cultured in explant medium (DMEM supplemented with 10% FBS, 400 μM NaOH, 1% PSG, and 1 mM luciferin, pH 7.2). Dishes were covered with 40 mm glass coverslips (Fisher Scientific) and sealed with vacuum grease to prevent evaporation. The luminescence signals of reporter cells were monitored every 10 min using a LumiCycle luminometer (Actimetrics) at 37 °C without added $CO_2$. The circadian period was calculated using the LumiCycle software (Actimetrics). Results shown are representative of at least three independent experiments.

**Western blotting**. Frozen mouse liver tissue was homogenized in RIPA buffer containing 1× EDTA-free protease inhibitor cocktail (Roche) by using Omni Tissue Homogenizer (Omni International). The concentration of total protein was determined by Bio-Rad Protein Assay and then equalized to 15 μg/μl. 25 μg of total protein was used for western blot assay which was performed as previously described[24]. Antibodies used in the western blots are anti-BMAL1 (Cell signaling, #14020, 1:1000), anti-HNF4A (Abcam, ab181604, 1:1000), and anti-TUBULIN (Sigma-Aldrich, T0198, 1:1000).

**Quantitative RT-PCR**. Liver tissues of 10–12-week-old male mice were harvested at indicated Zeitgeber times. Total RNA was isolated using TRIzol Reagent according to manufacturer's instructions (Life Technologies cat. #15596026), and then reverse transcribed to cDNA using iScript cDNA Synthesis kit (Bio-Rad cat. #1708891). We designed real-time primers spanning the exon–intron junctions using the IDT primer-designing software PrimerQuest (https://www.idtdna.com/PrimerQuest). Primer sequences are:

 *mRplp0*: forward GGCCCTGCACTCTCGCTTTC,
 reverse TGCCAGGACGCGCTTGT;
 *mHnf4a*: forward GTTCTGTCCCAGCAGATCAC,
 reverse GCTCCTTCATAGACTCACACAC;
 *mPer2*: forward GAGTGTGTGCAGCGGCTTAG,
 reverse GTAGGGTGTCATGCGGAAGG;
 *mDbp*: forward AATGACCTTTGAACCTGATCCCGCT,
 reverse GCTCCAGTACTTCTCATCCTTCTGT;
 *mNr1d1*: forward AGAGATGCTGTGCGTTTTGG,
 reverse AGGCTGCTCAGTTGGTTGTT;
 *mNr1d2*: forward AGTGGCATGGTTCTACTGTGT,
 reverse GCTCCTCCGAAAGAAACCCTTA;
 *mCry1*: forward GGGCTGGATCCACCATTTAG,
 reverse TCAAAGACCTTCATCCCTTCTTC;
 *mCry2*: forward GATGGAGGTTCCTACTGCAATC,
 reverse CAGCCTTGGGAACACATCA;
 *mBmal1*: forward CCCTAGGCCTTCATTGGATTT,
 reverse GCAAAGGGCCACTGTAGTT;
 *mClock*: forward CAGAACAGTACCCAGAGTGCT,
 reverse CACCACCTGACCCATAAGCAT.
 *hRplp0*: forward CGTGGAAGTGACATCGTCTT,
 reverse GGATGATCTTAAGGAAGTAGTTGGA;
 *hHnf4a*: forward TCTTTGACCCAGATGCCAAG,
 reverse GTCGTTGATGTAGTCCTCCAAG;
 *hBmal1*: forward ATCCTCAACTACAGCCAGAATG,
 reverse AGAGCTGCTCCTTGACTTTG;

RT-qPCR analyses were performed with CFX384 Real-Time PCR Detection System (Bio-Rad).

**ChIP-seq experiments**. ChIP-seq experiments with mouse liver were performed as previously described[24]. Briefly, blood perfused liver tissues were dissected and snap-frozen in liquid nitrogen. Frozen liver tissues were minced with razor blades and homogenized by pushing through 18 G needle followed by 21 G needle to release nuclei. Nuclei were immediately cross-linked with 1% formaldehyde for 10 min at room temperature and then quenched with glycine. After two washes, the nuclei were lysed in nuclear lysis buffer (50 mM Tris–HCl pH 8.0, 1% SDS, 10 mM EDTA) containing 1× EDTA-free protease inhibitor cocktail (Roche) and kept on ice for 5 min. Immediately, the lysates were sonicated 17 times for 30 s by using Bioruptor (Diagenode). After centrifugation at 17,000×g for 15 min at 4 °C, fragmented chromatin was diluted 10-fold with IP dilution buffer (20 mM Tris–HCl pH 8.0, 150 mM NaCl, 1% Triton X-100, 1 mM EDTA) supplemented with 1× EDTA-free protease inhibitor cocktail, and incubated with antibody overnight at 4 °C. Dynabeads Protein G (Invitrogen cat. #10004D) was added to the chromatin and incubated for another 3 h. Afterwards, the beads were washed once with low salt wash buffer (20 mM Tris–HCl pH 8.0, 150 mM NaCl, 1% Triton X-100, 0.1% SDS, 2 mM EDTA) and once with high salt wash buffer (20 mM Tris–HCl pH 8.0, 500 mM NaCl, 1% Triton X-100, 0.1% SDS, 2 mM EDTA). IP elution buffer (1% SDS, 100 mM NaHCO₃) was applied to the bead pellet and incubated at 30 °C for 15 min. The eluate was added with NaCl to the final conc. of 200 mM and reverse-crosslinked at 65 °C overnight. To remove RNA and protein, RNase A and proteinase K were subsequently applied by incubating at 45 °C for 1 h. Finally, DNA was purified with QIAquick PCR Purification columns.

ChIP-seq experiments with human cell lines were similar. The differences are the cells were crosslinked in culture dishes with 1% formaldehyde, and then scraped off for cell lysis and chromatin fragmentation. Antibodies used in the ChIP-seq experiments are anti-HNF4A (Abcam, ab41898), anti-BMAL1 (Cell Signaling, #14020), anti-H3K4me1 (Abcam, ab8895), anti-H3K27ac (Abcam, ab4729), and anti-FOXA2 (Abcam, ab256493).

**ChIP-seq analysis**. Single-end ChIP-seq reads were trimmed using Trimmomatic (v0.36) and then aligned to hg38 or mm10 genome with Bowtie2 (v2.3.4.1). BAM files were processed using SAMtools (v1.10) and PCR duplicates were removed with PicardTools (v2.18.3). Peaks were called in MACS2 (v2.1.2) using default settings and IgG mock ChIP files for normalization. BAM files of replicate samples were merged using SAMtools. BIGWIG track coverage files were generated from merged BAM files using the DeepTools (v3.3.0) bamCoverage command with RPGC normalization.

Heatmaps and metaplots were generated by the computeMatrix, plotHeatmap, and plotProfiles functions of DeepTools (v3.3.0) using BIGWIG files (replicates merged) and scaled regions. DiffBind (v3.2.7) was used to make PCA plots. Statistically significantly differential peaks were called and MA plots were generated by using the DESeq2 method within DiffBind, which selected differential regions based on ChIP signals in each replicate and FDR-corrected q-value of 0.05.

HOMER (v4.11.1) mergePeaks program was used to identify overlapping binding loci of two transcription factors. In order to define the sites as "overlapping", peak centers of the two binding sites must be at a distance ≤500 bp. Note that the peak numbers may not add up exactly since the function automatically resolves redundant overlaps by dropping one fragment during analysis. Motif enrichment analysis was performed using HOMER findMotifsGenome.pl command and scanned ± 200 bp from the peak center for binding sites of transcription factors, and ± 750 bp for histone modifications. HOMER annotatePeaks.pl command was used to make annotations of genomic features. The functional analyses of GO term ("Biological Process" sub-ontology) and KEGG pathway were performed using the clusterProfiler package in R or DAVID (https://david.ncifcrf.gov).

**ATAC-seq experiments**. The ATAC-seq procedure was based on a published method[68]. In brief, liver tissues were pulverized with mortar and pestle in liquid nitrogen and followed by nuclei permeabilization, tagmentation, library preparation, and Illumina HiSeq paired-end sequencing.

**ATAC-seq analysis**. Paired-end ATAC-seq reads were trimmed using Trimmomatic (v0.36) and then mapped to mm10 mouse genome using Bowtie2 (v2.3.4.1). SAMtools (v1.10) was used to generate BAM files, remove PCR duplicates, and remove mitochondrial DNA. MACS2 version 2.1.2 was used for peak calling with the following parameters: --nomodel --broad --shift -100 --extsize 200 --keep-dup all.

**Quantification and statistical analysis**. The significance of differences between peak distance, period length, and gene expression was evaluated by unpaired Student's $t$-test (two-tailed), with significant differences at $p < 0.05$. For motif analysis, HOMER findMotifsGenome.pl calculated $p$-values using the cumulative binomial distribution. For GO term and KEGG pathway analyses, clusterProfiler calculated $p$-values using a hypergeometric distribution which was then adjusted for multiple comparison.

**Reporting summary**. Further information on research design is available in the Nature Research Reporting Summary linked to this article.

## Data availability
The data that support this study are available from the corresponding authors upon reasonable request. Raw data (fastq files) and final processed data (bigWig and peak files) for NGS experiments are available on GEO under accession code GSE157452. GSE35262 and E-MTAB-941 were used to analyze PPARA, HNF1A, and LXR deposition at BMAL1 binding sites. GSE39860 and SRA025656 were used for reanalysis of H3K4me1 circadian rhythms. CircaDB [http://circadb.hogeneschlab.org/] was used for identification of circadian transcripts. Source data are provided with this paper.

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

## Acknowledgements

We thank Alexander Vu and other Kay lab members for technical support. This work was supported by the National Institute of Diabetes and Digestive and Kidney Diseases Grant 5R01DK108087 (to S.A.K.).

## Author contributions

M.Q. designed the study, performed data collection and bioinformatic analyses, and wrote the manuscript. H.Q. and Z.J. performed the majority of bioinformatic and statistical analyses. S.A.K. advised on study design, supported manuscript preparation, and supervised the project.

## Competing interests

The authors declare no competing interests.
