## [Peer Review File · Nature Communications]

REVIEWER COMMENTS

Reviewer #1 (Remarks to the Author):

In " HNF4A defines tissue-specific circadian 1 rhythms by beaconing BMAL1::CLOCK chromatin binding and shaping rhythmic chromatin landscape", Qu et al. describe how HNF4a may alter BMAL1 recruitment by functioning as a pioneer transcription factor. Rhythmicity of clock protein occupancy and the circadian nature of specific chromatin marks in the liver has been extensively studied. However, this study attempts to further understand how tissue-specific factors might contribute to clock (in this case, BMAL1, specifically) chromatin recruitment and activity. The authors provide interesting data involving the potential disruptive effects of a MODY1 mutation in HNF4a on hepatic rhythmicity.

Though a well-written paper and interesting topic, the choice of model raises many question as to the mechanisms underlying HNF4a's circadian activity, particularly as it relates to BMAL1:CLOCK recruitment and functon(details outlined below).

1. The liver-specific HNF4a knockout mouse has been previously reported to die at an early age due to severe metabolic defects, reaching >70% mortality by 8 weeks of age (PMID 11158324). Thus, based on the model, it is unclear whether the "fetal" vs. the adult form of HNF4a is ultimately affecting BMAL1 binding. In addition, the liver of this mouse is in a metabolically deteriorating state. Thus, it unclear the extent to which indirect effects on BMAL1 recruitment are present in this model. At least some isoform-specific knockdown experiments would be helpful to differentiate between isoform-specific function on circadian chromatin regulation during development. In addition, the HNF4a constitutive liver knockout has dramatic changes in PPAR α and LXR α expression. The expression and potential involvement of these transcription factors should be addressed experimentally.
2. Though HNF4a loss has been reported to result in dramatically different genomic profiles in female vs. male mice (PMID 18276827), this is not mentioned anywhere in the manuscript. Some indication of how HNF4a affects BMAL1 binding potentially in a sex-specific way should be addressed experimentally.
3. There is a noticeable absence of attention to the two isoforms of HNF4a. This is especially important as the fetal form directly represses Bmal1 transcriptionally (PMID 30341289). Though the BMAL1 protein does not appear to be dramatically changed in this model according to this manuscript, there may be adaptation throughout development since this is a global knockout. This isoform-specific transcriptional repression should be discussed at a minimum. This should also be discussed in the context of Figure 3, where overexpression of "HNF4a" (which isoform?) results in a slight reduction in Bmal1 expression. The two isoforms should be compared for these in vitro experiments.
4. Hep3B and HepG2 cells are both cancer lines that express both isoforms of HNF4a, with partial cytoplasmic localization of the adult isoform. Again, these experiments comparing the U2OS and liver cancer BMAL1 peaks are confusing. It is assumed that the authors are introducing the adult form into the U2OS cells, but then they are going to be comparing the binding to varying isoforms in the cancer lines. While their binding sites do overlap somewhat, the actual effects on gene expression can be different. See preprint doi.org/10.1101/2021.02.28.433261. This should be discussed at a minimum.
5. Authors have previously published that HNF4a has a repressive effect on several of the clock genes. The authors should discuss more directly the dampening effect of HNF4a loss of Per2, Dbp, and Nr1d1 rhythms in the HKO (Fig. 4). There could be indirect mechanisms responsible for this finding, considering HNF4a has been reported to repress in this context. I suppose that the lack of BMAL1 recruitment is what could be responsible, though PER2 is strongly regulated by CREB and does not necessarily lack rhythmicity in the absence of BMAL1. Can the authors please discuss these findings in further detail?
6. Can the authors please comment and compare their findings to those in PMID22936566, where the H3k4me1 mark appears to be highest at ZT5.8 in the liver.

7. It may be a typographical mistake, but in the discussion, the authors state that HNF4a is only expressed in liver and kidney. It is also expressed in the intestine (PMID 2279702 and other refs).

Reviewer #2 (Remarks to the Author):

General comments: In this manuscript, the authors have extended their previous studies showing that HNF4a is a circadian regulator in the liver. In these extensive studies, mainly in liver tissue, the authors show that BMAL1 chromatin loading is HNF4a dependent and is associated with open state of chromatin marked by H3K27ac and H3K4me1. The studies are well performed but a major question about the age of mice used arises (see below). Further, the manuscript lacks connecting the molecular events of TF binding to chromatin with function at either gene or organ level. This is critical, because these studies raise the question whether there is any functional consequence of having a HNF4a-BMAL1 loop regulating genes in the context of liver pathobiological events where HNF4a is of critical importance such as liver regeneration, liver cirrhosis, liver cancer and NAFLD.

Detailed comments:

1. The introduction is a bit rambling and contains information of histone changes which is otherwise well known. More focus on how circadian rhythm plays a role in liver biology should be included after introducing the HNF4a angle. This is critically important as this investigation is primarily in the liver and HNF4a is essential for its function. Further, HNF4a has different roles in different organs. The introduction gives an impression that what is discovered in the liver also applies to other organs, which may not be true.
2. It is surprising that the investigators could use HNF4a-KO mice at the age of 10-12 weeks. When HNF4a-KO mice are generated by breeding HNF4a-floxed mice with Albumin-cre mice, the resulting KO mice do not survive beyond 6-7 weeks (PMID: 11158324/PMCID: PMC99591 and PMID: 12808453). This is because loss of HNF4a during postnatal growth period results in failure to differentiate the liver leading to organ failure and death.
3. The authors present data showing that HNF4a drive BMAL1 binding to chromatin, but not on all genes. What is the biological consequence of HNF4a-dependent BMAL1 loading? What functions do the genes that show HNF4a-dependent BMAL1 loading do?
4. What are the genes that lose HNF4a binding in BKO? What are their functions?
5. What is the mechanism by which HNF4a induces H3K27ac and H3K4me1 that leads to BMAL1 loading?

Reviewer #3 (Remarks to the Author):

The manuscript by Qu et al. reports a tour-de-force study documenting the role of Hnf4a as a plausible pioneering factor in mouse liver-specific expression of circadian transcripts under the control of Bmal1-Clock. Using loss and gain-of-function analysis of genetically engineered mouse models with loss Hnf4a or Bmal1, the authors found that Hnf4 loss led to diminished Bmal1 chromatin binding associated with loss of enhancer markers, H3K4me1 and H3K27ac. Conversely, loss of Bmal1 diminished Hnf4a mediated transcription, suggesting that Bmal1 is required for full expression of 'rush hour' genes at late night and early morning. Overall, the study is compelling and makes use of clean genetic tools to provide a conceptual framework of how tissue selective transcription factors may influence circadian transcriptional output.

The manuscript provides a fundamental understanding of tissue-selective circadian transcription and would benefit general readers if the author would provide additional insights through gene ontology analysis of the transcriptomic alterations. For example, for the circadian transcripts that are most affected by loss of Hnf4 or Bmal1 (ie, overlap), it could be instructive to know the possible cellular

functions of these genes in terms of organ selectivity.

It is notable the Hnf4a transcript still peaked at ZT8-12 with loss of Bmal1 (Figure 6d). The author could provide a comment on this observation regarding whether feeding could affect Hnf4a circadian expression.

RESPONSE: We appreciate the reviewers for their enthusiasm and constructive comments on our manuscript. We have materially addressed ALL their questions, comments, and suggestions by incorporating **24 panels of entirely new data** and written information into the revised manuscript. We believe that these revisions not only address the reviewers' concerns but also provide further strong support for our findings described in the initial submission. Please find below our point-by-point response to the reviewers' comments.

Reviewer #1 (Remarks to the Author):

In "HNF4A defines tissue-specific circadian 1 rhythms by beaconing BMAL1::CLOCK chromatin binding and shaping rhythmic chromatin landscape", Qu et al. describe how HNF4a may alter BMAL1 recruitment by functioning as a pioneer transcription factor. Rhythmicity of clock protein occupancy and the circadian nature of specific chromatin marks in the liver has been extensively studied. However, this study attempts to further understand how tissue-specific factors might contribute to clock (in this case, BMAL1, specifically) chromatin recruitment and activity. The authors provide interesting data involving the potential disruptive effects of a MODY1 mutation in HNF4a on hepatic rhythmicity.

Though a well-written paper and interesting topic, the choice of model raises many question as to the mechanisms underlying HNF4a's circadian activity, particularly as it relates to BMAL1:CLOCK recruitment and functon(details outlined below).

RESPONSE: We thank the reviewer for noting that our results are novel and interesting. We appreciate that revisions were required to clarify several key points of the manuscript. We have substantially revised the manuscript throughout and have provided extensive new data, analyses, and written clarification to support our claims in the manuscript.

Reviewer 1, detailed comment 1a: The liver-specific HNF4a knockout mouse has been previously reported to die at an early age due to severe metabolic defects, reaching >70% mortality by 8 weeks of age (PMID 11158324). Thus, based on the model, it is unclear whether the "fetal" vs. the adult form of HNF4a is ultimately affecting BMAL1 binding. At least some isoform-specific knockdown experiments would be helpful to differentiate between isoform-specific function on circadian chromatin regulation during development.

RESPONSE: We appreciate the reviewer's question. In fact, due to spatiotemporal expression of Albumin in hepatocytes, the Cre activity in Albumin-Cre transgenic mice does not achieve efficient recombination of the loxP sequences until 6-weeks of age (PMID 10686614). Therefore, we would like to note that the liver-specific *Hnf4a* knockout mice used in this study may represent an animal model primarily for the adult form of HNF4A. Indeed, *Hnf4a* flox mice were crossed with *Alfp-Cre* mice in which the albumin promoter is controlled by the *Afp* enhancer for studying HNF4A roles in the fetal liver (PMID: 12808453). Echoing our mouse studies, the U2OS-based assays in our original manuscript indicated that expressing the adult isoform HNF4A2 was sufficient to induce BMAL1 binding. The reviewer has asked an interesting question as to whether the fetal form shares a similar activity. While several recent publications discovering novel hepatic functions of HNF4A have exclusively investigated the adult form (PMID: 32770044; 30933372; 33078654), in response to

reviewer's question, we cloned and stably expressed the fetal isoform HNF4A8 in U2OS cells and then performed ChIP-seq experiments for BMAL1 protein.

We found that ectopically expressing HNF4A8 suppressed *Bmal1* transcription (**Response Fig. 1a**) and remarkably induced tissue-specific BMAL1 bindings relative to the GFP expression control (**Response Fig. 1b**). Like we have described for HNF4A2 in Fig. 3, the HNF4A8-induced BMAL1 peaks were more likely located at distal or intronic enhancer regions (**Response Fig. 1c**) and enriched with the E-box element and HNF4A-binding motif (**Response Fig. 1d**). BMAL1 and HNF4A ChIP signals in the liver cancer cells are more abundant at the U2OS-HNF4A8-induced BMAL1 peaks (**Response Fig. 1e**), indicating the HNF4A8-induced BMAL1 binding events may mirror the in vivo pattern present in liver cells in situ. Remarkably, comparison of the HNF4A2- and HNF4A8-induced BMAL1 binding sites indicated a considerable overlap (**Response Fig. 1f-g**). Given that the exact roles of the fetal HNF4As in liver development are still poorly understood and their expression tends to be induced in liver diseases, these new results provide valuable insights arguing that HNF4A-regulated BMAL1 recruitment is invariable during liver development and disease transition.

Response Figure 1 (Revised Supplementary Fig. 6). The fetal isoform HNF4A8 is capable of inducing BMAL1 genome binding. (a) Transcript level of genes in U2OS-GFP or U2OS-HNF4A8 cells was determined by RT-qPCR. Displayed are the means \pm SD ($n = 3$) normalized to *Rplp0* expression levels. Statistical significance was determined by Student's t-test (** $P < 0.01$, *** $P < 0.001$). (b) MA plot showing differential BMAL1 occupancy in U2OS-GFP and U2OS-HNF4A8 cells, using threshold of FDR < 0.1 . The x-axis represents the mean number of reads (log scaled) within the peaks across all samples. The y-axis represents the log fold change between the two samples. (c) Distribution of genomic annotations of HNF4A8-enhanced BMAL1 peaks. (d) Motif analysis of HNF4A8-enhanced BMAL1 binding sites. *de novo* consensus motifs are shown with corresponding enrichment significance values. (e) BMAL1 peaks in U2OS-GFP and U2OS-HNF4A8 cells were partitioned into three categories with DiffBind. Then the corresponding BMAL1 and HNF4A occupancy in Hep3B or HepG2 cells were plotted by centering at each BMAL1 binding site in U2OS cells. Each horizontal line represents a single BMAL1 binding site in

U2OS. Peaks were ordered vertically by strength of BMAL1 ChIP signal in U2OS. (f) Venn diagram showing overlap between BMAL1-binding sites that were induced by HNF4A2 or HNF4A8 in U2OS cells. (g) IGV genome tracks showing BMAL1 and HNF4A enrichment at the *SLC25A42*, *DOK4*, *CDHR2*, and *PLPP3* gene loci in the indicated cells, based on normalized ChIP-seq read coverage. Track heights are indicated.

Reviewer 1, detailed comment 1b: In addition, the liver of this mouse is in a metabolically deteriorating state. Thus, it unclear the extent to which indirect effects on BMAL1 recruitment are present in this model. In addition, the HNF4a constitutive liver knockout has dramatic changes in PPARa and LXRA expression. The expression and potential involvement of these transcription factors should be addressed experimentally.

RESPONSE: We appreciate the suggestion by the reviewer. Based on the recommendations, we measured expression levels of PPARa, LXRA, and the well-characterized HNF4A target transcription factor HNF1A in the *Hnf4a* knockout liver, confirming that they were all downregulated (**Response Fig. 2a**). To address the reviewer's concern, we interrogated DNA sequences enriched at the HKO-reduced BMAL1 binding sites, finding that the E-box and HNF4A binding motifs were most enriched whereas the binding motifs for PPARa, LXRA, and HNF1A ranked far lower (**Response Fig. 2b**). Further, we analyzed the legacy PPARa, LXR, and HNF1A ChIP-seq data of mouse liver (PMID: 22158963; 22780989), discovering that these HNF4A-regulated TFs were equally distributed at the HKO-unchanged and HKO-reduced BMAL1 binding sites, standing in stark contrast to the differential occupancy of HNF4A (**Response Fig. 2c**). Collectively, these results strongly argue that the HNF4A-dependent BMAL1 binding events are specifically associated with HNF4A co-occupancy and therefore HNF4A directly regulates BMAL1 recruitment.

Response Figure 2 (Revised Fig. 1e and Supplementary Fig. 1a, 2d). HNF4A may directly induce BMAL1 chromatin binding. (a) Transcript level of genes was determined by RT-qPCR using liver samples isolated from control or HKO mice at ZT18. Displayed are the means \pm SD ($n = 4$) normalized to *Rplp0* expression levels. Statistical significance was determined by Student's t-test (** $P < 0.01$, *** $P < 0.001$). (b) Motif analysis of HKO-depleted BMAL1 binding sites. Known consensus motifs are shown with corresponding enrichment significance values. (c) BMAL1 peaks in control and HKO livers were partitioned into three categories with DiffBind (the HKO-enriched group has only 3 peaks and couldn't be plotted), and then the corresponding TF occupancy at each BMAL1 binding site was plotted. Each horizontal line represents a single BMAL1 binding site. Peaks were ordered vertically by strength of BMAL1 ChIP signal in control liver.

Reviewer 1, detailed comment 2: Though HNF4a loss has been reported to result in dramatically different genomic profiles in female vs. male mice (PMID 18276827), this is not mentioned anywhere in the manuscript. Some indication of how HNF4a affects BMAL1 binding potentially in a sex-specific way should be addressed experimentally.

RESPONSE: We thank the reviewer for the question. As the reviewer pointed out, even though the rate of HCC development induced by high fat diet was sex-independent in liver-specific *Hnf4a* knockout mice (PMID 31575546), the *Hnf4a* knockout liver exhibited more severe pathological lesions and greater changes in gene expression in male mice than the female (PMID 11158324; 18276827). Having taken these observations into account and to eliminate sex as a confounding factor, as stated in the original Methods, we have used male mice only throughout the study. While of interest (and has been discussed in the revised manuscript, **page 6, line 12-15**), we respectfully suggest that further exploration of the sex-specific HNF4A regulation of BMAL1 binding is beyond the scope of the present study.

Reviewer 1, detailed comment 3: There is a noticeable absence of attention to the two isoforms of HNF4a. This is especially important as the fetal form directly represses *Bmal1* transcriptionally (PMID 30341289). Though the BMAL1 protein does not appear to be dramatically changed in this model according to this manuscript, there may be adaptation throughout development since this is a global knockout. This isoform-specific transcriptional repression should be discussed at a minimum. This should also be discussed in the context of Figure 3, where overexpression of “HNF4a” (which isoform?) results in a slight reduction in *Bmal1* expression. The two isoforms should be compared for these in vitro experiments.

RESPONSE: We thank the reviewer for the suggestions. In response, we cloned and stably expressed the fetal isoform HNF4A8 in U2OS cells, and then performed RT-qPCR and ChIP-seq experiments for BMAL1. **Response Fig. 3a** demonstrates that HNF4A8 reduced *Bmal1* transcription, although no greater than HNF4A2 did (Fig. 3b). Considering that the ectopic expression level of HNF4A8 was lower than that of HNF4A2 (by ~2 folds), potentially due to lower integrated lentivirus copy number or faster mRNA turnover, our results may be not contradictory to the observations in PMID 30341289. Regarding the liver-specific *Hnf4a* knockout mice, as we have addressed in response to reviewer’s comment 1a, they may represent a knockout model specifically for the adult form of HNF4A. This may provide additional explanation why we did not see a dramatic change in *Bmal1* expression. As suggested by the reviewer, we have discussed the isoform-specific *Bmal1* transcriptional regulation in the revised manuscript (**page 20, line 14-17**). In addition, we demonstrate in the BMAL1 ChIP-seq experiment that, like HNF4A2, the fetal form HNF4A8 was capable of inducing BMAL1 relocalization to tissue-specific binding sites (**Response Fig. 3b-g**).

Response Figure 3 (Revised Supplementary Fig. 6). The fetal isoform HNF4A8 is capable of inducing BMAL1 genome binding. (a) Transcript level of genes in U2OS-GFP or U2OS-HNF4A8 cells was determined by RT-qPCR. Displayed are the means \pm SD ($n = 3$) normalized to *Rplp0* expression levels. Statistical significance was determined by Student's t-test (** $P < 0.01$, *** $P < 0.001$). (b) MA plot showing differential BMAL1 occupancy in U2OS-GFP and U2OS-HNF4A8 cells, using threshold of FDR < 0.1 . The x-axis represents the mean number of reads (log scaled) within the peaks across all samples. The y-axis represents the log fold change between the two samples. (c) Distribution of genomic annotations of HNF4A8-enhanced BMAL1 peaks. (d) Motif analysis of HNF4A8-enhanced BMAL1 binding sites. *de novo* consensus motifs are shown with corresponding enrichment significance values. (e) BMAL1 peaks in U2OS-GFP and U2OS-HNF4A8 cells were partitioned into three categories with DiffBind. Then the corresponding BMAL1 and HNF4A occupancy in Hep3B or HepG2 cells were plotted by centering at each BMAL1 binding site in U2OS cells. Each horizontal line represents a single BMAL1 binding site in

U2OS. Peaks were ordered vertically by strength of BMAL1 ChIP signal in U2OS. (f) Venn diagram showing overlap between BMAL1-binding sites that were induced by HNF4A2 or HNF4A8 in U2OS cells. (g) IGV genome tracks showing BMAL1 and HNF4A enrichment at the *SLC25A42*, *DOK4*, *CDHR2*, and *PLPP3* gene loci in the indicated cells, based on normalized ChIP-seq read coverage. Track heights are indicated.

Reviewer 1, detailed comment 4: Hep3B and HepG2 cells are both cancer lines that express both isoforms of HNF4a, with partial cytoplasmic localization of the adult isoform. Again, these experiments comparing the U2OS and liver cancer BMAL1 peaks are confusing. It is assumed that the authors are introducing the adult form into the U2OS cells, but then they are going to be comparing the binding to varying isoforms in the cancer lines. While their binding sites do overlap somewhat, the actual effects on gene expression can be different. See preprint doi.org/10.1101/2021.02.28.433261. This should be discussed at a minimum.

RESPONSE: We thank the reviewer for this question. In the preprint paper, the authors found the genome-wide localizations of the fetal and adult HNF4As considerably overlap, with ~96% of the binding sites being indeed identical. This observation makes a lot of sense considering that the two isoforms share exactly the same DNA-binding domain and are different only in the N-terminal activation domain AF-1. Besides, our new results indicate that both isoforms are capable of inducing tissue-specific BMAL1 genome binding (**Response Fig. 1 and 3**). Taking these facts into account, we did not discriminate between the two forms in the liver cell-based HNF4A ChIP-seq experiments. These experiments are crucial to demonstrate that the BMAL1 binding events that we have artificially induced in U2OS cells by HNF4A expression largely mirror the in vivo pattern present in liver cells in situ. Since the liver cell-based datasets were exclusively used for genome binding analyses, we respectfully suggest that the isoform-specific transcriptional activity should not be a consideration here. As suggested by the reviewer, we have discussed this issue in the revised manuscript (**page 12, line 1-3**).

Reviewer 1, detailed comment 5: Authors have previously published that HNF4a has a repressive effect on several of the clock genes. The authors should discuss more directly the dampening effect of HNF4a loss of *Per2*, *Dbp*, and *Nr1d1* rhythms in the HKO (Fig. 4). There could be indirect mechanisms responsible for this finding, considering HNF4a has been reported to repress in this context. I suppose that the lack of BMAL1 recruitment is what could be responsible, though *PER2* is strongly regulated by CREB and does not necessarily lack rhythmicity in the absence of BMAL1. Can the authors please discuss these findings in further detail?

RESPONSE: We thank the reviewer for the comments and totally agree that reduced BMAL1 recruitment is largely responsible for the dampened expression of clock genes in the HKO liver. To address the reviewer's question, we would like to note that the trans-repression by HNF4A requires BMAL1 to engage the target genes beforehand. However, BMAL1 recruitment was substantially attenuated by *Hnf4a* knockout at those clock genes. Therefore, rather than a relief from HNF4A repression, reduced BMAL1 recruitment is the primary effect of *Hnf4a* knockout, leading to downregulation of the BMAL1-CLOCK target genes. Based on this question, we have further explained the mechanisms of dampened clock genes in the revised Discussion (**page 21, line 2-7**).

Reviewer 1, detailed comment 6: Can the authors please comment and compare their findings to those in PMID 22936566, where the H3k4me1 mark appears to be highest at ZT5.8 in the liver.

RESPONSE: We thank the reviewer for raising this question. It should be noted that Fig. 4B of PMID 22936566 displays H3K4me1 deposition exclusively at the TSS regions. To interrogate H3K4me1 occupancy on a genome-wide scale, which is more relevant to our analysis in Fig. 5a, we analyzed H3K4me1 datasets generated in the paper (GSE39860). **Response Fig. 4a** shows that the genome-wide deposition of H3K4me1 peaked around CT8 and troughed around CT12. In the meanwhile, we identified another study that characterized H3K4me1 circadian rhythms in the liver (PMID 23217262). By analyzing the deposited datasets (SRA025656), we found the genome-wide deposition of H3K4me1 reached its highest level during the night (**Response Fig. 4b**), which is indeed consistent with our observation. Although the H3K4me1 enrichment peak time are not quite the same for the three studies, it was commonly observed that the genome-wide H3K4me1 deposition started to rise after dark (CT12), coinciding with the increased recruitment of HNF4A. As suggested by the reviewer, we have discussed these relevant prior studies in the revised manuscript (**page 21, line 9-11**).

Response Figure 4 (Revised Supplementary Fig. 9g, 9h). Circadian rhythms of H3K4me1 deposition characterized in literature PMID 22936566 (a) and PMID 23217262 (b). Profiles of genome-wide H3K4me1 distribution throughout the day were plotted using DeepTools (v3.3.0).

Reviewer 1, detailed comment 7: It may be a typographical mistake, but in the discussion, the authors state that HNF4a is only expressed in liver and kidney. It is also expressed in the intestine (PMID 2279702 and other refs).

RESPONSE: We apologize for the confusion. Indeed, the statement was made while we discuss a transcriptome study that investigated 12 mouse organs not including the intestine (PMID: 25349387). Based on the question, we have more clearly explained the context of the statement in the revised Discussion (**page 21, line 17-18**) and cited the important literature in the revised Introduction (**page 5, line 1-2**).

Reviewer #2 (Remarks to the Author):

General comments: In this manuscript, the authors have extended their previous studies showing that HNF4a is a circadian regulator in the liver. In these extensive studies, mainly in liver tissue, the authors show that BMAL1 chromatin loading is HNF4a dependent and is associated with open state of chromatin marked by H3K27ac and H3K4me1. The studies are well performed but a major question about the age of mice used arises (see below). Further, the manuscript lacks connecting the molecular events of TF binding to chromatin with function at either gene or organ level. This is critical, because these studies raise the question whether there is any functional consequence of having a

HNF4a-BMAL1 loop regulating genes in the context of liver pathobiological events where HNF4a is of critical importance such as liver regeneration, liver cirrhosis, liver cancer and NAFLD.

RESPONSE: We thank the reviewer for the comments on our manuscript and appreciate the recognition that our studies are extensive and well performed. In response to the reviewer's questions, we have substantially revised the manuscript by providing new data, analyses, and written information regarding how the regulation of BMAL1 binding by HNF4A relates to normal liver functions as well as pathophysiology.

Reviewer 2, detailed comment 1: The introduction is a bit rambling and contains information of histone changes which is otherwise well known. More focus on how circadian rhythm plays a role in liver biology should be included after introducing the HNF4a angle. This is critically important as this investigation is primarily in the liver and HNF4a is essential for its function. Further, HNF4a has different roles in different organs. The introduction gives an impression that what is discovered in the liver also applies to other organs, which may not be true.

RESPONSE: We thank the reviewer for the comments. As suggested, we have substantially revised the manuscript Introduction (**page 3-5**) to reflect the critical roles of the circadian clock in liver biology and minimize the background information related to pioneer TF and chromatin remodeling. The section now focuses on our current research in the liver and has avoided further predicting HNF4A roles in other organs.

Reviewer 2, detailed comment 2: It is surprising that the investigators could use HNF4a-KO mice at the age of 10-12 weeks. When HNF4a-KO mice are generated by breeding HNF4a-floxed mice with Albumin-cre mice, the resulting KO mice do not survive beyond 6-7 weeks (PMID: 11158324/PMCID: PMC99591 and PMID: 12808453). This is because loss of HNF4a during postnatal growth period results in failure to differentiate the liver leading to organ failure and death.

RESPONSE: We appreciate the opportunity to clarify this key point. While the higher survival rate of our liver-specific HKO mice was likewise surprising to us, indeed, these mice live to at least the age of 9 months. Also, all alleles were inherited in a Mendelian fashion, indicating that the *Hnf4a* knockout did not result in prenatal lethality. To interpret this distinction, we would like to note that we have crossed the *Hnf4a* flox mice (JAX#004665) with Alb-Cre mice (JAX#003574) and Per2-luciferase circadian rhythm reporter mice (JAX#006852) constructed in the same C57BL/6J background, whereas the Hayhurst HKO model (PMID: 11158324) has used Alb-Cre mice generated in the background of FVB strain (JAX#016833). It seems that different genetics contributed by the distinct Alb-Cre lines has caused the strain-specific lethality, given that the genetic background does make a difference when analyzing mouse phenotypes (PMID: 32284572; <https://www.jax.org/news-and-insights/2006/june/the-importance-of-genetic-background-in-mouse-based-biomedical-research>).

To address the reviewer's concern, we characterized our liver-specific *Hnf4a* knockout mice at the molecular, cellular and tissue levels. By the age of 10-11 weeks, *Hnf4a* expression was reduced by ~75% in the HKO liver, which was further confirmed by downregulation of the classic HNF4A target genes *ApoC3*, *Fabp1*, *Ppara*, and *Hnf1a* (**Response Fig. 5a, 5b**). The liver-to-body-weight ratio was significantly increased for the HKO mice (**Response Fig. 5c**). The HKO hepatocytes exhibited marked vacuolization giving them an "empty" appearance (**Response Fig. 5d**). Moreover, the HKO

liver was pale in appearance and exhibited a significant increase in lipid accumulation as measured by Oil Red O staining (**Response Fig. 5d**). Given that these phenotypes all match typical outcomes of *Hnf4a* loss, we reason that our mice have better tolerated the pathological lesions and therefore will serve as a valuable resource for studying long-term effects of *Hnf4a* knockout in the adult liver. In the revised manuscript, we have included these new data and discussed the prior animal models (page 5, line 3-6; page 6, line 1-12).

Response Figure 5 (Revised Fig. 1d and Supplementary Fig. 1). Characterization of liver-specific *Hnf4a* knockout mice (*Hnf4a*^{fl/fl} *Alb-Cre*^{+/-} *Per2-luc*^{+/+}) at 10-11 weeks of age. (a) Transcript level of genes was determined by RT-qPCR using liver samples isolated from control or HKO mice at ZT18. Displayed are the means \pm SD (n = 4) normalized to *Rplp0* expression levels. Statistical significance was determined by Student's t-test (**P < 0.01, ***P < 0.001). (b) Western blot analysis using liver samples isolated from control or HKO mice at ZT6. (c) Liver to body weight ratio of the control or HKO liver (n = 4). Statistical significance was determined by Student's t-test (**P < 0.01). (d) Histopathological analysis of the *Hnf4a* knockout livers. Representative images of H&E staining (upper panel) and Oil Red O staining (lower panel). Paraffin-embedded liver sections were used for H&E staining and frozen sections from the same samples were used for Oil Red O staining. All images are 400X.

Reviewer 2, detailed comment 3: The authors present data showing that HNF4a drive BMAL1 binding to chromatin, but not on all genes. What is the biological consequence of HNF4a-dependent BMAL1 loading? What functions do the genes that show HNF4a-dependent BMAL1 loading do?

RESPONSE: We thank the reviewer for raising the question. The list of genes that lost BMAL1 binding in the HKO liver has been attached to the revised manuscript (**Revised Supplementary Table 1**). To address the reviewer's question, first, we show in original Fig. 1b and 4 that BMAL1 occupancy at the core circadian clock genes was substantially reduced by *Hnf4a* knockout, leading to dampened gene expression and circadian rhythm disruption. To understand the functions of the other

HNF4A-dependent BMAL1 binding genes, we performed KEGG and gene ontology (GO) pathway enrichment analyses for the complete set of HKO-reduced BMAL1 binding sites (**Response Fig. 6**). In addition to the circadian clock machinery, the HKO-reduced BMAL1 binding genes were highly enriched in metabolic pathways, such as glucose and cholesterol metabolism, especially when compared with the unchanged BMAL1 binding sites. This finding argues that the circadian regulation of these key tissue-specific nodes is supervised by HNF4A.

Response Figure 6 (Revised Supplementary Fig. 2b, 2c). Functional analysis of the HKO-reduced BMAL1 binding genes. (a) KEGG pathway enrichment analysis of the HKO-unchanged or HKO-reduced BMAL1 binding genes. (b) Gene ontology (GO) (“biological process” sub-ontology) terms associated with the HKO-reduced BMAL1 binding sites.

Reviewer 2, detailed comment 4: What are the genes that lose HNF4a binding in BKO? What are their functions?

RESPONSE: We thank the reviewer for the questions. In response, we have provided the list of genes that lost HNF4A binding in the BKO liver (**Revised Supplementary Table 3**). Further, we performed KEGG and GO term pathway analyses for these genes, revealing genes involved in cancer pathogenesis to be one of the most enriched gene groups (**Response Fig. 7**). Other categories found significantly overrepresented included genes involved in the Wnt/ β -catenin signaling pathway and the cell cycle. HNF4A was characterized to inhibit the Wnt/ β -catenin signaling pathway and cell cycle progression, potentially underlying its tumor suppressive roles (PMID: 33462379). These functional analyses therefore indicate that the most critical aspect of HNF4A action in the liver is modulated by the circadian clock.

Response Figure 7 (Revised Supplementary Fig. 11c, 11d). Functional analysis of the BKO-reduced HNF4A binding genes. (a) KEGG pathway enrichment analysis of the BKO-unchanged or BKO-reduced HNF4A binding genes. (b) Gene ontology (GO) (“biological process” sub-ontology) terms associated with the BKO-reduced HNF4A binding sites.

Reviewer 2, detailed comment 5: What is the mechanism by which HNF4a induces H3K27ac and H3K4me1 that leads to BMAL1 loading?

RESPONSE: We thank the reviewer for the question, which remains an unresolved mystery for most cases of pioneer TF regulation. One case that has been explored in mechanistic depth is the FOXA family of pioneer factors. FOXA binding was able to independently displace core histones from local chromatin as assayed with a reconstituted nucleosomal array, relying on the “winged helix” DNA-binding domain structure that highly resembles and competes with that of the linker histone H1 (PMID: 27058788). However, this mechanism does not seem to apply to other pioneer TFs which very likely act by recruiting ATP-dependent chromatin remodelers (PMID: 30675018; 29507097). It will be of interest to employ direct nucleosome binding assays to determine how HNF4A DNA-binding domain may adapt to the nucleosome surface and whether it may engage ATP-dependent enzymes to expand the “openness” of a local chromatin domain. Given the brief period permitted by Nature Communications for revision, we will pursue these efforts in future studies.

Despite the different mechanisms underlying pioneer factor-induced chromatin opening, histone modifications H3K4me1 and H3K27ac were commonly found associated with pioneer binding and chromatin accessibility. In other words, as the reviewer remarked in the beginning, H3K27ac and H3K4me1 mark the open state of chromatin. Besides, H3K4me1 has an active regulatory role in chromatin opening by serving as a docking site for chromatin remodelers (PMID: 29255264; 33712604). H3K27ac is recognized by BRD4, which recruits Mediator and RNA Pol II to establish enhancer-promoter interactions and activate cell type-specific gene expression (PMID: 29263365). Prior studies collectively suggest that the pioneer binding induces sequential deposition of the histone

modifications by recruiting histone H3K4 methyltransferases MLL3/4 and H3K27 acetyltransferases CBP/p300 (PMID: 27926873, 24368734; 28398509). In response to the reviewer's question, we have provided comments in the revised Introduction and Discussion (**page 4, line 5-11; page 19, line 6-9**).

Reviewer #3 (Remarks to the Author):

The manuscript by Qu et al. reports a tour-de-force study documenting the role of Hnf4a as a plausible pioneering factor in mouse liver-specific expression of circadian transcripts under the control of Bmal1-Clock. Using loss and gain-of-function analysis of genetically engineered mouse models with loss Hnf4a or Bmal1, the authors found that Hnf4 loss led to diminished Bmal1 chromatin binding associated with loss of enhancer markers, H3K4me1 and H3K27ac. Conversely, loss of Bmal1 diminished Hnf4a mediated transcription, suggesting that Bmal1 is required for full expression of 'rush hour' genes at late night and early morning. Overall, the study is compelling and makes use of clean genetic tools to provide a conceptual framework of how tissue selective transcription factors may influence circadian transcriptional output.

RESPONSE: We thank the reviewer for the positive evaluation of our manuscript.

Reviewer 3, detailed comment 1: The manuscript provides a fundamental understanding of tissue-selective circadian transcription and would benefit general readers if the author would provide additional insights through gene ontology analysis of the transcriptomic alterations. For example, for the circadian transcripts that are most affected by loss of Hnf4 or Bmal1 (ie, overlap), it could be instructive to know the possible cellular functions of these genes in terms of organ selectivity.

RESPONSE: We thank the reviewer for the insightful comment and suggestion. In response, we performed circadian rhythm analysis for 319 genes that were significantly downregulated by liver-specific *Hnf4a* knockout (PMID 23315968) and 462 genes that were most differentially expressed at all time points by *Bmal1* knockout in the liver (PMID 26843191). By querying the circadian transcriptional profile database CircaDB, we found that 38% (122/319) of the HNF4A-downregulated transcripts and 37% (171/462) of the BMAL1-regulated transcripts were robustly rhythmic (**Revised Supplementary Table 2**), higher than the ratio of ~16% for general hepatic transcripts (PMID: 25349387). Moreover, phase distribution of the BMAL1- and HNF4A-regulated circadian transcripts indicates they both tend to peak at the pre-dawn "rush hours" (**Response Fig. 8a, 8b**). Gene ontology (GO) ("biological process" sub-ontology) (**Response Fig. 8c, 8d**) and KEGG pathway (**Response Fig. 8e, 8f**) functional enrichment analyses revealed the BMAL1- or HNF4A-regulated circadian genes to be similarly involved in circadian rhythm regulation and critical aspects of hepatic functions, *i.e.* lipid and cholesterol metabolism, amino acid metabolism, redox reactions, and liver development. These new analyses provide strong supports for our hypothesis that HNF4A plays critical roles in the regulation of tissue-specific circadian rhythms and are incorporated into the revised manuscript (**page 15, line 5-14**).

c Circadian genes differentially expressed in *Bmal1* knockout liver

GO term	# Genes	FDR
lipid metabolic process	15	4.10E-02
triglyceride metabolic process	5	4.10E-02
cholesterol metabolic process	7	4.10E-02
fatty acid metabolic process	8	9.00E-02
ADP biosynthetic process	3	9.00E-02
oxidation-reduction process	16	9.60E-02
glutathione metabolic process	5	9.60E-02
liver development	6	1.30E-01
nucleobase-containing compound metabolic process	4	2.10E-01
cellular response to lithium ion	3	5.30E-01
negative regulation of cell-cell adhesion	3	5.60E-01
L-cysteine catabolic process	2	8.40E-01
response to insulin	4	9.90E-01
circadian sleep/wake cycle	2	9.90E-01
response to cadmium ion	3	9.90E-01

d Circadian genes downregulated in *Hnf4a* knockout liver

GO term	# Genes	FDR
lipid metabolic process	14	2.40E-03
steroid metabolic process	7	3.20E-03
cholesterol metabolic process	7	3.20E-03
oxidation-reduction process	14	2.60E-02
circadian rhythm	5	3.00E-01
fatty acid metabolic process	5	8.00E-01
response to interleukin-18	2	8.70E-01
acute-phase response	3	8.70E-01
liver development	4	8.70E-01
triglyceride metabolic process	3	9.00E-01
kynurenine metabolic process	2	9.00E-01
negative regulation of peptidase activity	4	9.20E-01
amyloid precursor protein catabolic process	2	9.20E-01
L-serine metabolic process	2	9.20E-01
tryptophan catabolic process to kynurenine	2	9.60E-01

e Circadian genes differentially expressed in *Bmal1* knockout liver

KEGG pathway	# Genes	FDR
Glyoxylate and dicarboxylate metabolism	6	1.50E-03
Metabolic pathways	31	4.10E-03
Glycerolipid metabolism	5	1.80E-01
Carbon metabolism	6	2.30E-01
Biosynthesis of amino acids	5	2.30E-01
PPAR signaling pathway	5	2.40E-01
Peroxisome	5	2.40E-01
Tryptophan metabolism	4	2.40E-01
alpha-Linolenic acid metabolism	3	4.40E-01
Drug metabolism - cytochrome P450	4	4.60E-01
Fat digestion and absorption	3	7.40E-01

f Circadian genes downregulated in *Hnf4a* knockout liver

KEGG pathway	# Genes	FDR
Metabolic pathways	27	1.60E-04
Steroid biosynthesis	4	2.30E-02
Tryptophan metabolism	5	2.30E-02
PPAR signaling pathway	4	8.70E-01
Fat digestion and absorption	3	1.00E+00
Glycerolipid metabolism	3	1.00E+00
Synthesis and degradation of ketone bodies	2	1.00E+00
Hepatitis C	4	1.00E+00

Response Figure 8 (Revised Supplementary Fig. 10). Transcripts most altered in *Hnf4a* or *Bmal1* knockout liver tend to be rhythmically expressed. (a-b) Phase distribution of circadian transcripts most altered in *Bmal1* (a) or *Hnf4a* (b) knockout liver. (c-d) GO terms (“biological process” sub-ontology) associated with circadian transcripts most altered in *Bmal1* (c) or *Hnf4a* (d) knockout liver were determined by DAVID (<https://david.ncifcrf.gov/home.jsp>). (e-f) KEGG functional pathways associated with circadian transcripts most altered in *Bmal1* (e) or *Hnf4a* (f) knockout liver were determined by DAVID.

Reviewer 3, detailed comment 2: It is notable the *Hnf4a* transcript still peaked at ZT8-12 with loss of

Bmal1 (Figure 6d). The author could provide a comment on this observation regarding whether feeding could affect *Hnf4a* circadian expression.

RESPONSE: We appreciate the recommendation. In response, we first measured mRNA levels of the classic circadian genes *Dbp* and *Per2*, showing that the *Dbp* oscillation was substantially abolished while the *Per2* transcripts were still cycling in the *Bmal1* knockout liver (**Response Fig. 9a**). The preserved *Per2* oscillation under dysfunctional hepatic clock has been previously described, likely attributable to feeding-induced changes in CREB and heat shock factor (HSF) activities (PMID: 17298173; 19940241). In contrast, we found in Qu *et al.* 2018 (PMID 30530698) that the night-enhanced *Hnf4a* expression was not altered by fasting (**Response Fig. 9b**). Therefore, *Hnf4a* oscillation may employ a clock-independent mechanism distinct from *Per2* cycling. As suggested by the reviewer, we have commented on these observations in the revised manuscript (**page 16-17**).

Response Figure 9 (Revised Fig. 6d and PMID 30530698 Fig. S6). Feeding effects on clock-independent *Hnf4a* oscillation. (a) Control and BKO mouse livers were harvested at 4-h intervals over the course of 24 h. Transcript level of genes was analyzed by using RT-qPCR. Displayed are the means \pm SD ($n = 3$) normalized to non-oscillating *Rplp0* expression levels. Statistical significance was determined by Student's t-test (* $P < 0.05$, ** $P < 0.01$, *** $P < 0.001$). (b) Mice were fed normally or fasted from ZT4 to ZT14. Liver tissues were collected at ZT14 and transcript level of genes were quantified by RT-qPCR. Displayed are the means \pm SD ($n = 5$) normalized to non-oscillating *Rplp0* expression levels. Statistical significance was determined by Student's t-test (* $P < 0.05$, ** $P < 0.01$, *** $P < 0.001$).

REVIEWERS' COMMENTS

Reviewer #1 (Remarks to the Author):

The authors have performed a number of key experiments that help with interpretation of the previous results. My main concerns were related to the model system, but the authors have been able to logically explain possible reasons for the increase in lifespan. Importantly, they have added such discussion to the manuscript. I am glad that the authors were able to interrogate additional datasets related to the H3K4me1 marks. It appears that there is some disparity in the literature. I am satisfied with other responses by the authors.

Reviewer #2 (Remarks to the Author):

The authors have greatly improved the manuscript based on the comments and have added substantial new data. The fact that the HNF4alpha-KO mice can live beyond first couple of months is intriguing and should be highlighted. The background variation as an explanation is good at present but there is certainly some compensation from other nuclear factors, especially CEBPalpha, that may be involved. While this doesn't change the interpretation of the data at present, this should be brought up in the discussion.